# FONDUE: an algorithm to automatically find the dimensionality of the latent representations of variational autoencoders

## Abstract

When training a variational autoencoder (VAE) on a given dataset, determining the number of latent variables requires human supervision and to fully train one or more VAEs. In this paper, we explore ways to simplify this time-consuming process through the lens of the polarised regime. Specifically, we show that the discrepancies between the variance of the mean and sampled representations of a VAE reveal the presence of passive variables in the latent space, which, in well-behaved VAEs, indicates a superfluous number of dimensions. After formally demonstrating this phenomenon, we use it to propose FONDUE: an unsupervised algorithm which efficiently finds a suitable number of latent dimensions without fully training any models. We additionally show that FONDUE can be extended in a number of ways, providing a principled and unified method for selecting the number of latent dimensions for VAEs and deterministic autoencoders.

## 1 Introduction

"How many latent variables should I use for this model?" is the question that many practitioners using variational autoencoders (VAEs) or autoencoders (AEs) have to deal with. When the task has been studied before, this information is available in the literature for the specific architecture and dataset used. However, when it has not, answering this question becomes more complicated. Indeed, the current methods to estimate the latent dimensions require human supervision or at least one fully trained model (Doersch, 2016; Mai Ngoc & Hwang, 2020; Yu & Príncipe, 2019; Boquet et al., 2021). Moreover, most of the proposed solutions are not theoretically grounded.

One could wonder if, instead of looking for an appropriate number of dimensions, it would be sufficient to use a very large number of latent dimensions in all cases. However, beside defeating the purpose of learning compressed representations, this may lead to a range of issues. For example, one would obtain lower accuracy on downstream tasks (Mai Ngoc & Hwang, 2020) and—if the number of dimensions is sufficiently large—very high reconstruction loss (Doersch, 2016). This would also hinder the interpretability of downstream task models such as linear regression, prevent investigating the learned representation with latent traversal (Locatello et al., 2019b), and increase the correlation between the latent variables (Bonheme & Grzes, 2021).

VAEs with isotropic Gaussian priors have been shown to learn in a polarised regime (Dai & Wipf, 2018; Rolinek et al., 2019; Bonheme & Grzes, 2023) where superfluous dimensions—the passive variables—are "ignored" by the decoder during reconstruction. This regime leads to discrepancies between the mean and sampled representations, which are indicative of the amount of passive variables present in the latent representation (Bonheme & Grzes, 2021). Thus, one can reasonably assume that a "good" number of latent dimensions corresponds to the maximum number of active variables (i.e., variables used by the decoder for reconstruction) a model can learn before the appearance of passive variables.

In this paper, we demonstrate that the amount of passive variables present in a latent representation can easily be detected by analysing the discrepancies between the mean and sampled representations. Specifically, we

show that one can bound the difference between the traces of the covariance matrices of both representations. Based on this property, we then design a simple yet efficient unsupervised algorithm which fulfills the criteria of the current methods (i.e., low reconstruction loss, high accuracy on downstream tasks, high compression, etc.), without requiring to fully train any models.

Our contributions are as follows:

(i) We prove that the discrepancies between the mean and sampled representations of well-behaved VAEs can be used to monitor the type of variables (active or passive) learned by a VAE.

(ii) Based on this theoretical insight, we propose FONDUE: an algorithm which finds the number of latent dimensions that leads to a low reconstruction loss and good accuracy. In opposition to current methods (Doersch, 2016; Mai Ngoc & Hwang, 2020; Yu & Príncipe, 2019; Boquet et al., 2021), it does not require human supervision or to fully train multiple models.

(iii) The library created for this experiment is available in supplemental work and can be reused with other models or techniques (see paragraph on extending FONDUE of Section 3) for further research in the domain. A notebook illustrating the usage of FONDUE can also be found at `https://anonymous.4open.science/r/fondue-demo-CD5A`.

**Notational considerations**  Throughout this paper, we will use the superscript $^{(i)}$ to denote the values obtained for the i$^{th}$ sample $\mathbf{x}^{(i)}$ of the random variable $\mathbf{x}$, and represent the j$^{th}$ dimension of a vector representation using the subscript $_j$. Similarly, given a subset of indexes $\boldsymbol{a} = 1, \cdots, k$, the superscript $^{(\boldsymbol{a})}$ will denote the values obtained for the samples $\mathbf{x}^{(1,\ldots,k)}$ of the random variable $\mathbf{x}$, and the subscript $_{\boldsymbol{a}}$ will represent the dimensions $1, ..., k$ of a vector representation. As shown in Figure 1, we will also use a shortened version of the mean, variance and sampled representations, such that $\boldsymbol{\mu} \triangleq \boldsymbol{\mu}(\mathbf{x}; \boldsymbol{\phi})$, $\boldsymbol{\sigma} \triangleq diag[\boldsymbol{\Sigma}(\mathbf{x}; \boldsymbol{\phi})]$, and $\mathbf{z} \triangleq \boldsymbol{\mu} + \boldsymbol{\epsilon}\boldsymbol{\sigma}^{1/2}$, respectively. Here $\boldsymbol{\epsilon}$ is a noise variable such that $\boldsymbol{\epsilon} \sim \mathcal{N}(\mathbf{0}, \boldsymbol{I})$, $\boldsymbol{\phi}$ denotes the parameters of the encoder, and the diag$[\cdot]$ operator returns the diagonal values of a matrix. For example, $\boldsymbol{\sigma} = \text{diag}[\boldsymbol{\Sigma}(\mathbf{x}; \phi)]$ is the vector of variance values obtained from the diagonal of the covariance matrix $\boldsymbol{\Sigma}(\mathbf{x}; \phi)$. Similarly, for a specific data example $\mathbf{x}^{(i)}$ and noise sample $\boldsymbol{\epsilon}^{(i)}$, $\boldsymbol{\mu}^{(i)} \triangleq \boldsymbol{\mu}(\mathbf{x}^{(i)}; \phi)$, $\boldsymbol{\sigma}^{(i)} \triangleq diag[\boldsymbol{\Sigma}(\mathbf{x}^{(i)}; \phi)]$ and $\mathbf{z}^{(i)} \triangleq \boldsymbol{\mu}^{(i)} + \boldsymbol{\epsilon}^{(i)}(\boldsymbol{\sigma}^{(i)})^{1/2}$. For multiple data examples $\boldsymbol{X} = \{\mathbf{x}^{(i)}\}_{i=0}^{h}$, $\boldsymbol{M} \triangleq [\boldsymbol{\mu}^{(0)} \cdots \boldsymbol{\mu}^{(h)}]^T$, $\boldsymbol{S} \triangleq [\boldsymbol{\sigma}^{(0)} \cdots \boldsymbol{\sigma}^{(h)}]^T$, $\boldsymbol{E} \triangleq [\boldsymbol{\epsilon}^{(0)} \cdots \boldsymbol{\epsilon}^{(h)}]^T$, and $\boldsymbol{Z} \triangleq [\mathbf{z}^{(0)} \cdots \mathbf{z}^{(h)}]^T$.

## 2 Background

### 2.1 Intrinsic dimension estimation

It is generally assumed that a dataset $\boldsymbol{X} = \{\mathbf{x}^{(i)}\}_{i=0}^{h}$ of $h$ i.i.d. data examples $\mathbf{x}^{(i)} \in \mathbb{R}^m$ is a locally smooth non-linear transformation $g$ of a lower-dimensional dataset $\boldsymbol{Y} = \{\mathbf{y}^{(i)}\}_{i=0}^{h}$ of $h$ i.i.d. samples $\mathbf{y}^{(i)} \in \mathbb{R}^d$, where $d \leqslant m$ (Campadelli et al., 2015; Chollet, 2021). The goal of Intrinsic dimension (ID) estimation is to recover $d$ given $\boldsymbol{X}$. In recent years, these techniques have successfully been applied to deep learning to empirically show that the intrinsic dimension of images was much lower than their extrinsic dimension (i.e., the number of pixels) (Gong et al., 2019; Ansuini et al., 2019; Pope et al., 2021). Based on these findings, we will use Intrinsic Dimension Estimates (IDEs) as the initial number of dimensions $n$ for FONDUE in Section 3.2. In practice, we compute the IDEs using two ID estimation techniques: MLE (Levina & Bickel, 2004) and TwoNN (Facco et al., 2017). See Appendix E and Campadelli et al. (2015) for more details on ID estimation techniques.

### 2.2 Non-Variational Autoencoders

Deep deterministic autoencoders (AEs) (Kramer, 1991) can be thought of as a non-linear version of PCA (Baldi & Hornik, 1989). They are composed of an encoder $f_{\boldsymbol{\phi}}(\mathbf{x})$ which maps an input $\mathbf{x}$ to a compressed representation $\mathbf{z}$, and a decoder $g_{\boldsymbol{\theta}}(\mathbf{z})$ which attempts to reconstruct $\mathbf{x}$ from a compressed representation $\mathbf{z}$. AEs are optimised to minimise the reconstruction error $\mathcal{L}(\mathbf{x}, g_{\boldsymbol{\theta}}(\mathbf{z}))$ (e.g., MSE).

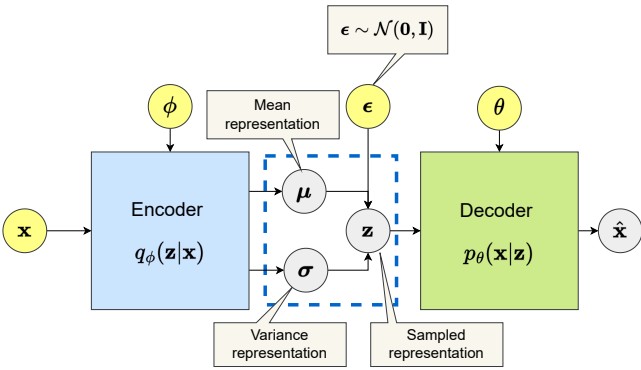

Figure 1: Illustration of a VAE. The distributions are generally assumed to be multivariate Gaussian, $\boldsymbol{\mu}$ is the mean representation and $\boldsymbol{\sigma}$ is the variance representation. $\boldsymbol{\mu}$ and $\boldsymbol{\sigma}$ are the parameters of the posterior over $\mathbf{z}$.

## 2.3 Variational Autoencoders

VAEs (Kingma & Welling, 2014; Rezende & Mohamed, 2015) are deep probabilistic generative models based on variational inference. As illustrated in Figure 1, the encoder maps an input $\mathbf{x}$ to a latent representation $\mathbf{z}$, and the decoder attempts to reconstruct $\mathbf{x}$ using $\mathbf{z}$. This can be optimised by maximising $\mathcal{L}$, the evidence lower bound (ELBO)

$$\mathcal{L}(\boldsymbol{\theta}, \boldsymbol{\phi}; \mathbf{x}) = \underbrace{\mathbb{E}_{q_{\boldsymbol{\phi}}(\mathbf{z}|\mathbf{x})}[\log p_{\boldsymbol{\theta}}(\mathbf{x}|\mathbf{z})]}_{\text{reconstruction term}} - \underbrace{D_{\mathrm{KL}}\big(q_{\boldsymbol{\phi}}(\mathbf{z}|\mathbf{x}) \,\|\, p(\mathbf{z})\big)}_{\text{regularisation term}}, \tag{1}$$

where $p(\mathbf{z})$ is generally modelled as a standard multivariate Gaussian distribution $\mathcal{N}(\mathbf{0}, \boldsymbol{I})$ to permit a closed form computation of the regularisation term (Doersch, 2016). $q_{\boldsymbol{\phi}}(\mathbf{z}|\mathbf{x})$ and $p_{\boldsymbol{\theta}}(\mathbf{x}|\mathbf{z})$ are also commonly assumed to be Gaussians (Dai et al., 2017; Dai & Wipf, 2018). The regularisation term can be further penalised by a weight $\beta$ (Higgins et al., 2017) such that

$$\mathcal{L}(\boldsymbol{\theta}, \boldsymbol{\phi}; \mathbf{x}) = \underbrace{\mathbb{E}_{q_{\boldsymbol{\phi}}(\mathbf{z}|\mathbf{x})}[\log p_{\boldsymbol{\theta}}(\mathbf{x}|\mathbf{z})]}_{\text{reconstruction term}} - \underbrace{\beta D_{\mathrm{KL}}\big(q_{\boldsymbol{\phi}}(\mathbf{z}|\mathbf{x}) \,\|\, p(\mathbf{z})\big)}_{\text{regularisation term}}, \tag{2}$$

reducing to equation 1 when $\beta = 1$ and to a deterministic autoencoder when $\beta = 0$ (where only the reconstruction loss is optimised).

## 2.4 Polarised regime

The polarised regime (a.k.a selective posterior collapse) is the ability of VAEs to "shut down" superfluous dimensions of their sampled latent representations $\mathbf{z}$ while providing a high precision on the remaining variables (Rolinek et al., 2019; Dai & Wipf, 2018; Dai et al., 2020). As a result, for each input $\mathbf{x}^{(i)}$, the sampled representation can be separated into two subsets of variables, active and passive. The active variables correspond to the subset of the latent representation that is needed for the reconstruction. In opposition, the passive variables do not have any influence on the decoder and are thus only optimised with respect to the KL divergence. Given a Gaussian decoder with diagonal covariance $\gamma \boldsymbol{\Sigma_\theta}$, Dai & Wipf (2018) proved that for any active variable $j$, $\lim_{\gamma \to 0} \mathbf{z}_j^{(i)} = \boldsymbol{\mu}_j^{(i)}$. In the same way, for any passive variable $j$, $\lim_{\gamma \to 0} \mathbf{z}^{(i)} = \boldsymbol{\epsilon}^{(i)}$. In practice, this limit is approached very early in the training (i.e., after a few epochs), as shown in Dai & Wipf (2018, Fig.1.a).

Bonheme & Grzes (2021) proposed an extension of the polarised regime for multiple data examples. They hypothesised that the polarised regime could lead to three cases: (1) a variable is active for all the data samples, (2) a variable is passive for all the data samples, (3) a variable is mixed: passive for some samples and active for others. This results in the following definition.

**Definition 1** (Polarised regime of $\boldsymbol{Z}$)**.** *When a VAE learns in a polarised regime, its sampled representation* $\boldsymbol{Z}$ *is composed of a set of passive, active and mixed variables* $\mathbb{V}_p \cup \mathbb{V}_a \cup \mathbb{V}_m$ *such that:*

*(i)* $\lim_{\gamma \to 0} \boldsymbol{Z}_j = \boldsymbol{E}_j \quad \forall\, j \in \mathbb{V}_p,$

*(ii)* $\lim_{\gamma \to 0} \boldsymbol{Z}_j = \boldsymbol{M}_j \quad \forall\, j \in \mathbb{V}_a,$

*(iii)* $\lim_{\gamma \to 0} \boldsymbol{Z}_j^{(\boldsymbol{p})} = \boldsymbol{E}_j^{(\boldsymbol{p})}$ *and* $\lim_{\gamma \to 0} \boldsymbol{Z}_j^{(\boldsymbol{a})} = \boldsymbol{M}_j^{(\boldsymbol{a})} \quad \forall\, j \in \mathbb{V}_m.$

They concluded that mixed and passive variables were responsible for any discrepancies between the mean and sampled representations. This finding will be the base of our analysis in Section 3.

**Assumptions** In the rest of this paper, we assume that the considered models are learning under the polarised regime, which naturally happens for VAEs whose prior is Gaussian with diagonal covariance (Rolinek et al., 2019; Bonheme & Grzes, 2023) and reasonable values of $\beta$ (i.e., when $\beta$ is not large enough to lead to posterior collapse). As discussed above, Dai & Wipf (2018) have also shown that active and passive variables can be observed very early in the training of such models (i.e., after the first few epochs) and we will see in Section 3 that this early convergence assumption plays an important role on the computational time of FONDUE. These requirements can be made without loss of generality as it has been shown that the polarised regime was necessary for VAEs to learn properly (Dai & Wipf, 2018; Dai et al., 2018; 2020) and happens very early in training (Dai & Wipf, 2018; He et al., 2019; Bonheme & Grzes, 2022).

## 2.5 Related work

To the best of our knowledge, the literature on finding an appropriate number of latent dimensions for VAEs is limited, and the existing techniques—mostly designed for deterministic autoencoders—generally rely on approaches requiring human supervision. A great majority of these techniques are based on the Elbow (a.k.a. scree plot) method (James et al., 2013) which visually finds the point where a curve "bends" before diminishing returns occur.

**Elbow method using the reconstruction error** Doersch (2016) trained multiple models with different numbers of latent dimensions and selected the ones with the lowest reconstruction error. They noted that models' performances were noticeably worse when using extreme numbers of latent dimensions $n$. In their experiment, this happened for $n < 4$ and $n > 10,000$ on MNIST.

**Elbow method using the accuracy on downstream tasks** Mai Ngoc & Hwang (2020) suggested to train multiple models with different numbers of latent dimensions, and then compare the accuracy of the latent representations on a downstream task. They observed that while a higher number of latent dimensions could lead to a lower reconstruction error, it generally caused instability on downstream tasks. They thus concluded that the best number of latent dimensions for VAEs should be the one with the smallest classification variance and the highest accuracy, and they obtained similar results for AEs.

**Automated evaluation based on information theory** Yu & Príncipe (2019) observed that, from the data preprocessing inequality, $\mathrm{I}(\mathbf{x}, \hat{\mathbf{x}}) \geqslant \mathrm{H}(\mathbf{z})$. From this, Boquet et al. (2021) proposed to automatically find the number of latent dimensions of deterministic autoencoders by comparing the values of $\mathrm{I}(\boldsymbol{X}, \hat{\boldsymbol{X}})$ and $\mathrm{H}(\boldsymbol{Z})$. While no formal proof was provided, they hypothesised that given an optimal number of dimensions $n^*$, any number of latents $n < n^*$ would result in $\mathrm{I}(\mathbf{x}, \hat{\mathbf{x}}) > \mathrm{H}(\mathbf{z})$, and any $n \geqslant n^*$ would lead to $\mathrm{I}(\mathbf{x}, \hat{\mathbf{x}}) \approx \mathrm{H}(\mathbf{z})$. From this, they proposed to do a binary search over a sorted array of latent dimensions $\boldsymbol{n} = [1, ..., n]$ defined by the user. At each iteration, they trained a new autoencoder with a bottleneck size $\boldsymbol{n}_i$, estimated $\mathrm{I}(\mathbf{x}, \hat{\mathbf{x}})$ and $\mathrm{H}(\mathbf{z})$ for each batch, and averaged the results over each epoch. Each model was fully trained unless $\mathrm{I}(\mathbf{x}, \hat{\mathbf{x}}) \approx \mathrm{H}(\mathbf{z})$ (i.e., the algorithm requires at least one fully trained model). As Yu & Príncipe (2019)[1], they used a kernel estimator of the Rényi's $\alpha$-order entropy with an RBF kernel but with the order of $\alpha = 2$. We will refer to their algorithm as the Information Bottleneck (IB) algorithm in the rest of the paper.

---

[1]More details about Yu & Príncipe (2019) can be found in Appendix J

**Differences with our contribution**   The work the most closely related to ours is the IB algorithm of Boquet et al. (2021) which is the only one proposing some level of automation. However, our contribution differs from theirs in several aspects: 1) our approach is based on the polarised regime, not on information theory; 2) we provide a theoretical justification of our algorithm and a proof of convergence while this was left for future work in Boquet et al. (2021); 3) our solution is generally faster than the original implementation of the IB algorithm as we do not require to fully train any model; 4) the IB algorithm requires human supervision to select a range of likely latent dimensions which is not needed by FONDUE. We will further see in Section 5 that the number of dimensions obtained with FONDUE provides a better trade-off between reconstruction quality and accuracy on downstream task than IB. In other words, we will see that FONDUE provides results closer to Elbow methods than IB.

## 3   Mean and sampled representations under the polarised regime

As we have seen in Section 2.4, the polarised regime leads to discrepancies between mean and sampled representations (Locatello et al., 2019a) which can be indicative of the type of variables learned by a VAE (Bonheme & Grzes, 2021). The aim of this section is to provide some theoretical insight into how the different types of variables impact the difference between the two representations.

Specifically, we will study the difference between the traces of the covariance matrices of both representations when the latent variables are composed of a combination of different types of variables. This will allow us to use bounded quantities to detect when passive variables start appearing. From this, we will show that the bounds of this difference are specific to the types of variables present in the latent representations.

Finally, to illustrate a possible usage of these bounds, we will propose an algorithm for Facilely Obtaining the Number of Latent Dimensions by Unsupervised Estimation (FONDUE) which can quickly provide an estimate of the maximum number of latent dimensions that a VAE can have without containing any (unused) passive variables. The proofs of all the propositions and theorems of this section can be found in Appendix A, and the impact of FONDUE on the reconstruction and downstream task performance will further be studied in the next section.

### 3.1   Identifying the types of variables learned

In this section we will derive the difference between the traces of the covariance matrices of $M$ and $Z$ when the latent representation is composed of: 1) only active variables, 2) active and passive variables, 3) active and mixed variables. From these results we will then present the bounds of the difference between the traces of the covariance matrices of $M$ and $Z$ at $n+1$ latent variables when 1) $n$ is the maximal number of active variables, and 2) $n$ is the maximal number of non-passive variables. We will see in Section 3.2 that these two cases will inform the choice of the threshold value for FONDUE. The proofs of all the following propositions and theorems can be found in Appendix A.1.To make the distinction between the cross-covariance matrices and variance-covariance matrices clear, we will use $\mathrm{Cov}[\boldsymbol{A}, \boldsymbol{B}]$ for the former and $\mathrm{Var}[\boldsymbol{A}] \triangleq \mathrm{Cov}[\boldsymbol{A}, \boldsymbol{A}]$ for the latter.

First, let us consider the case where all variables are active. As mentioned in Section 2.4, in this case $\lim_{\gamma \to 0} \boldsymbol{Z} = \boldsymbol{M}$. Thus, Proposition 1 trivially follows:

**Proposition 1.** *If all the $n$ latent variables of a VAE are active, then* $\lim_{\gamma \to 0} \mathrm{Tr}(\mathrm{Var}[\boldsymbol{Z}]) - \mathrm{Tr}(\mathrm{Var}[\boldsymbol{M}]) = 0$.

When the latent representation is composed of active and passive variables only, one can decompose the variances into active and passive parts. Then, recalling that passive variables have a variance of 1 in sampled representations and close to 0 in mean representations, and using Proposition 1 again for the active part, it follows that:

**Proposition 2.** *If $s$ of the $n$ latent variables of a VAE are passive and the remaining $n-s$ variables are active, then* $\lim_{\gamma \to 0} \mathrm{Tr}(\mathrm{Var}[\boldsymbol{Z}]) - \mathrm{Tr}(\mathrm{Var}[\boldsymbol{M}]) = s$.

While mixed variables (i.e., variables that are passive for some input and active for others) are less trivial to analyse, using the same techniques as Proposition 2, and recalling that mixed variables come from a mixture distribution, we have:

**Proposition 3.** *If $s$ of the $n$ latent variables of a VAE are mixed and the remaining $n - s$ variables are active, then $0 < \lim_{\gamma \to 0} \mathrm{Tr}(\mathrm{Var}[\boldsymbol{Z}]) - \mathrm{Tr}(\mathrm{Var}[\boldsymbol{M}]) < s$.*

We are now interested in how the difference of the traces evolves when we increase the number of latent dimensions. When we have $n$ active variables, we know form Proposition 1 that the difference of trace will be close to 0. Using Propositions 2 and 3 with $s = 1$, we know that if the next variable is not active, the difference will increase by at most 1.

**Theorem 1.** *$n$ is the maximal number of dimensions for which a VAE contains only active variables if*

- *for $n$ latent variables, $\lim_{\gamma \to 0} \mathrm{Tr}(\mathrm{Var}[\boldsymbol{Z}]) - \mathrm{Tr}(\mathrm{Var}[\boldsymbol{M}]) = 0$,*

- *and for $n + 1$ latent variables, $0 < \lim_{\gamma \to 0} \mathrm{Tr}(\mathrm{Var}[\boldsymbol{Z}]) - \mathrm{Tr}(\mathrm{Var}[\boldsymbol{M}]) \leqslant 1$.*

When we have $n$ non-passive variables, using Proposition 3 with $s \leqslant n$, the difference of the trace will be between higher than 0 but lower than $n$. Using Proposition 2 with $s = 1$, we know that if the next variable is passive, the difference will increase by 1.

**Theorem 2.** *$n$ is the maximal number of dimensions for which a VAE contains only non-passive variables if*

- *for $n$ latent variables, $0 \leqslant \lim_{\gamma \to 0} \mathrm{Tr}(\mathrm{Var}[\boldsymbol{Z}]) - \mathrm{Tr}(\mathrm{Var}[\boldsymbol{M}]) < n$,*

- *and for $n + 1$ latent variables, $1 \leqslant \lim_{\gamma \to 0} \mathrm{Tr}(\mathrm{Var}[\boldsymbol{Z}]) - \mathrm{Tr}(\mathrm{Var}[\boldsymbol{M}]) < n + 1$.*

A useful property of Theorems 1 and 2 is that they hold very early in training. Indeed, the variance of the decoder very quickly approaches 0 (Dai & Wipf, 2018; Dai et al., 2018) leading to observations of the polarised regime after a few epochs (Rolinek et al., 2019; Bonheme & Grzes, 2022). This allows Theorems 1 and 2 to provide stable estimates using mean and sampled representations which reflect accurately the number of active and passive variables present in the final model. While these theorems can be useful by themselves to study the type of representations learned by an already trained VAE, we believe that they can also help to design new tools to improve models' quality. For example, by providing an estimation of the maximum number of non-passive latent dimensions that a VAE can reach for a given dataset. We will see in Section 3.2 that this can be done in an unsupervised way, without fully training any models or manually applying the Elbow method.

### 3.2 Finding the number of dimensions by unsupervised estimation

As discussed in Section 3.1, the traces of the covariance matrices of the mean and sampled representations start to diverge when non-active variables appear. We can thus use Theorems 1 and 2 to find the number of latent dimensions retaining the most information while remaining highly compressed (i.e., no passive variables).

For example, let us consider a threshold $t = 1$. We know from Theorem 1 that if the difference between traces is lower than or equal to $t$ for $n$ latent dimensions but higher than $t$ for $n+1$ latent dimensions, then $n$ is the largest number of dimensions for which the latent representation contains only active variables. If, on the other hand we allow $t$ to be higher, the model can encode additional non-active variables, as per Theorem 2. After selecting a suitable value for $t$ depending on the desired level of compression, we can thus iteratively check the difference between traces for different values of $n$ after training each model for a few steps until we obtain the largest $n$ for which the difference is lower than the threshold $t$ that we defined. To this aim, we propose an algorithm to Facilely Obtaining the Number of Latent Dimensions by Unsupervised Estimation (FONDUE).

**Theorem 3.** *Any execution of FONDUE (Algorithm 1) returns the largest number of dimensions $n$ for which $\mathrm{Tr}(\mathrm{Var}[\boldsymbol{Z}]) - \mathrm{Tr}(\mathrm{Var}[\boldsymbol{M}]) \leqslant t$:*

- *If $t < 1$, FONDUE returns the maximal number of latent dimensions for which a VAE contains only active variables.*

- *If $t \geqslant 1$, FONDUE returns the maximal number of latent dimensions for which a VAE contains only non-passive variables.*

*Sketch Proof.* In Algorithm 1, we define a lower and upper bound of $n$, $l$ and $u$, and update the predicted number of latent variables $n$ until, after $i$ iterations, $n_i = l_i$. Using the loop invariant $l_i \leqslant n_i \leqslant u_i$, we can show that the algorithm terminates when $l_i = n_i = \text{floor}\left(\frac{l_i + u_i}{2}\right)$, which can only be reached when $u_i = n_i + 1$, that is, when $n_i$ is the maximum number of latent dimensions for which we have $\text{Tr}(\text{Var}[\boldsymbol{Z}]) - \text{Tr}(\text{Var}[\boldsymbol{M}]) \leqslant t$. The threshold $t$ is determined according to Theorems 1 and 2, as discussed above. See Appendix A.6 for the full proof. □

**How does FONDUE work?** FONDUE will seek to reach the maximum number of dimensions for which the difference between $\text{Tr}(\text{Var}[\boldsymbol{Z}])$ and $\text{Tr}(\text{Var}[\boldsymbol{M}])$ is lower than or equal to the threshold $t$, as illustrated in Figure 2. The number of dimensions is first initialised to the IDE of the dataset $IDE_{data}$ to start from a reasonable number of latents. Note that a random initialisation will not impact the number of dimensions predicted by FONDUE but may slow the algorithm down if the value is very far from the predicted number of dimensions. Indeed, FONDUE will need more iterations to converge in that case. However, as shown in Appendix H, even with extreme initialisation values, FONDUE remains faster than fully training one model. After initialisation, at each iteration, FONDUE will train a VAE for a few epochs (generally just two) and retrieve the mean and sampled representations corresponding to 10,000 data examples. We then compute the traces of the covariance matrices of the mean and sampled representations (i.e., $\text{Tr}(\text{Var}[\boldsymbol{Z}])$ and $\text{Tr}(\text{Var}[\boldsymbol{M}])$) and the difference between them. If this difference is lower than or equal to the threshold, we set our lower bound to the current number of latent dimensions and train a VAE again with twice the number of latents, as illustrated in Figure 3. If the difference is higher than the threshold, we set the current number of latent dimensions to our upper bound and train a VAE again with half of the sum of the lower and upper bound, as illustrated in Figure 4. We iterate these two steps until our current number of latent dimensions is the largest possible dimensionality for which the difference is smaller than or equal to the threshold.

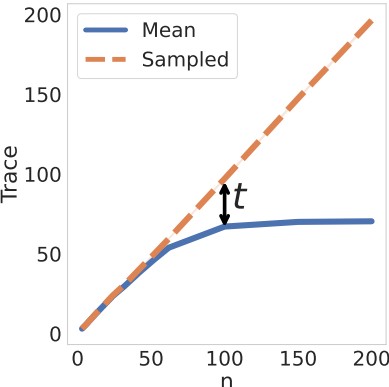

Figure 2: Traces of the covariance matrices of the mean and sampled representations of VAEs trained on Celeba with an increasing number of latent dimensions $n$. FONDUE retrieves the largest $n$ for which the difference is smaller or equal to a given threshold $t$.

**Extending FONDUE** Note that one could use different flavours of FONDUE by replacing the difference $\text{Tr}(\text{Var}[\boldsymbol{Z}]) - \text{Tr}(\text{Var}[\boldsymbol{M}])$ by any metric as long as its score is 1) consistently lower than some threshold $t$ before a "good" $n$ and higher or equal afterwards, 2) stable early in the training. One such example is $IDE_{\mathbf{z}} - IDE_{\boldsymbol{\mu}}$ which generally provides consistent results with the original algorithm as shown in Appendix F. We will see in Section 4.2 that the information theoretic metric of Boquet et al. (2021) can also be readily integrated into FONDUE.

**Algorithm 1** FONDUE

1: **procedure** FONDUE($t, IDE_{data}, e$)
2:     $l \leftarrow 0$                           ▷ Lower bound
3:     $u \leftarrow \infty$                          ▷ Upper bound
4:     $n \leftarrow IDE_{data}$     ▷ Current number of latent dimensions
5:     $m \leftarrow \{\}$
6:     **while** $n \neq l$ **do**
7:         $\mathrm{Tr}(\mathrm{Var}[\boldsymbol{Z}]), \mathrm{Tr}(\mathrm{Var}[\boldsymbol{M}]) \leftarrow$ GET-MEM($m, n, e$)
8:         **if** $\mathrm{Tr}(\mathrm{Var}[\boldsymbol{Z}]) - \mathrm{Tr}(\mathrm{Var}[\boldsymbol{M}]) \leqslant t$ **then**    ▷ Figure 3
9:             $l \leftarrow n$
10:           $n \leftarrow \min(n \times 2, u)$
11:         **else**                     ▷ Figure 4
12:             $u \leftarrow n$
13:             $n \leftarrow \mathrm{floor}\left(\frac{l+u}{2}\right)$
14:         **end if**
15:     **end while**
16:     **return** $n$
17: **end procedure**

**Algorithm 2** GET-MEM

1: **procedure** GET-MEM($m, n, e$)
2:     **if** $m[n] = \emptyset$ **then**
3:         $vae \leftarrow$ TRAIN-VAE($dim = n, n\_epochs = e$)
4:         $\mathrm{Tr}(\mathrm{Var}[\boldsymbol{Z}]), \mathrm{Tr}(\mathrm{Var}[\boldsymbol{M}]) \leftarrow Traces(vae)$
5:         $m[n] \leftarrow \mathrm{Tr}(\mathrm{Var}[\boldsymbol{Z}]), \mathrm{Tr}(\mathrm{Var}[\boldsymbol{M}])$
6:     **end if**
7:     **return** $m[n]$
8: **end procedure**

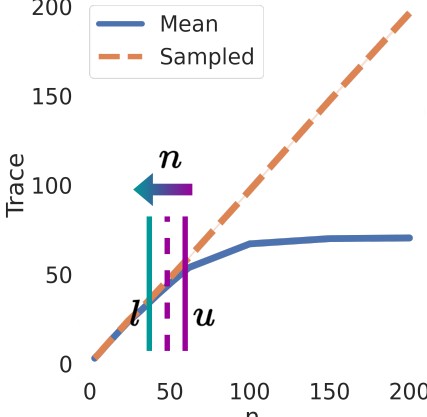

Figure 3: Update $l$ and increase $n$ until $\mathrm{Tr}(\mathrm{Var}[\boldsymbol{Z}]) - \mathrm{Tr}(\mathrm{Var}[\boldsymbol{M}]) > t$.

Figure 4: Update $u$ and decrease $n$ until $\mathrm{Tr}(\mathrm{Var}[\boldsymbol{Z}]) - \mathrm{Tr}(\mathrm{Var}[\boldsymbol{M}]) \leqslant t$.

## 4 Experimental setup

After computing the IDEs of the datasets on which the models will be trained, we will use them to initialise FONDUE. We will then assess the performance of FONDUE by comparing it with the existing techniques of dimension selection discussed in Section 2.5. This will be done by ensuring that the number of dimensions selected by the Elbow method for reconstruction and downstream tasks is consistent with the values proposed by FONDUE, and by comparing the results obtained with FONDUE and the IB algorithm of Boquet et al. (2021).

### 4.1 General setup

**Datasets**   We use three datasets with an increasing number of intrinsic dimensions: Symmetric solids (Murphy et al., 2021), dSprites (Higgins et al., 2017), and Celeba (Liu et al., 2015). The numbers of generative factors of the first two datasets are 2 and 5, respectively, and the IDE of these two datasets should be close to these values. While we do not know the generative factors of Celeba, Pope et al. (2021) reported an IDE of 26 when using $k = 20$ neighbours with MLE, indicating a higher number of intrinsic dimensions than dSprites and Symsol.

**Data preprocessing**   Each image is resized to $64 \times 64 \times c$, where $c = 1$ for Symmetric solids and dSprites, and $c = 3$ for Celeba. We also removed duplicate images (i.e., cases where different rotations resulted in the same image) and labels from Symmetric solids and created a reduced version: `symsol_reduced` which is available  at `https://t.ly/G7Qp`.

**VAE training**   We use the $\beta$-VAE architecture detailed in Higgins et al. (2017) for all the datasets, together with the standard learning objective of VAEs, as presented in Equation 1. Each VAE is trained 5 times with a number of latent dimensions $n = 3, 6, 8, 10, 12, 18, 24, 32, 42$ on every dataset. For Celeba, which has the highest IDE, we additionally train VAEs with latent dimensions $n = 52, 62, 100, 150, 200$.

**Estimations of the ID**   For all the datasets, we estimate the ID using 10,000 data examples. As in Pope et al. (2021), the MLE scores are computed with an increasing number of neighbours $k = 3, 5, 10, 20$. Moreover, we repeat the MLE computations 3 times with different seeds to detect any variance in estimates.

**Downstream tasks**   To monitor how good the learned latent representations are on downstream tasks, we train gradient boosted trees to classify the labels of each dataset based on the mean representations. Each label is learned as a separate classification task, and we evaluate the results on a dataset based on the averaged test accuracy over all these tasks, similarly to (Locatello et al., 2019a;b).

### 4.2   Comparing FONDUE with the IB algorithm

As the IB algorithm is the most closely related to FONDUE, we will compare the results of both algorithms. However, we will see that a few modifications to the IB algorithm are required to ensure a fair comparison in the unsupervised setting. For completeness, we also combine the results of Theorems 1 and 2 with binary search to compare with the IB algorithm in the supervised setting in Appendix K.

**Why the IB algorithm cannot be directly compared with FONDUE**   Because the IB algorithm performs binary search over a user-defined array of likely dimensions, its execution time is greatly dependent on the size of this array. Indeed, each time the dimension selected in the IB algorithm is lower than $n^*$, the corresponding model is trained until convergence. Thus, if we select a range of values from 1 to $n$ such that $floor(\frac{n-1}{2})$ is lower than $n^*$ we will always have at least one full model training, which, for the tested dataset is slower than FONDUE. For example, if $n^* = 30$, any $n < 61$ would trigger at least one full model training. Comparing the execution time of FONDUE and the original IB algorithm will thus mostly be based on whether the binary search encounters a situation where $n \geqslant n^*$ and require to fully train one or more models or not. Moreover, the original IB algorithm will give a value that depends on the range selected. For example, if $n^* = 30$ but the selected range is from 1 to 20, the original IB algorithm will return 20. Thus, manually selecting the results could also impact the quality of the predicted number of dimensions. For example one could force the IB algorithm to provide good predictions by selecting a very small range of values consistent with the Elbow methods while a larger range of values would provide worse predictions. To summarise, the human supervision plays an important role in the quality of the IB algorithm predictions, making it unpractical to compare with a fully unsupervised algorithm such as FONDUE. We thus propose to use a unified algorithm which does not require to define a range of values to circumvent this issue. As mentioned above, we also compare the original IB algorithm where all models are trained for a fixed number of epochs and a binary search for a fixed range of the number of dimensions using the difference between traces obtained in Theorem 3 in Appendix K. Appendix K is thus a comparison between the IB algorithm and Theorem 3 in a supervised context, where the range of dimensions to search is manually defined.

**Using FONDUE$_{IB}$ for a fair comparison**   The IB algorithm relies on the assumption that if the current number of dimensions $n$ is lower than the target number of dimensions $n^*$, $\mathrm{H}(\mathbf{z}) - \mathrm{I}(\mathbf{x}, \hat{\mathbf{x}}) < 0$ and $\mathrm{H}(\mathbf{z}) - \mathrm{I}(\mathbf{x}, \hat{\mathbf{x}}) \approx 0$ otherwise. One can thus directly use this inequality to provide an alternative version of FONDUE, FONDUE$_{IB}$, which will also be completely unsupervised and will not require to fully train one or more models. This is done by replacing $\mathrm{Tr}(\mathrm{Var}[\boldsymbol{Z}])$ and $\mathrm{Tr}(\mathrm{Var}[\boldsymbol{M}])$ by $\mathrm{H}(\mathbf{z})$ and $\mathrm{I}(\mathbf{x}, \hat{\mathbf{x}})$ in the original algorithm, as illustrated in Algorithms 3 and 4. In Boquet et al. (2021), the scores of $\mathrm{H}(\mathbf{z})$ and $\mathrm{I}(\mathbf{x}, \hat{\mathbf{x}})$ are truncated to 2 decimals, making $\mathrm{H}(\mathbf{z}) - \mathrm{I}(\mathbf{x}, \hat{\mathbf{x}}) < 0$ equivalent to $\mathrm{H}(\mathbf{z}) - \mathrm{I}(\mathbf{x}, \hat{\mathbf{x}}) \leqslant -0.01$. We thus set the

threshold to $t = -0.01$ for FONDUE$_{IB}$ and keep the original truncation. As in Boquet et al. (2021), we use an RBF kernel and set $\alpha = 2$.

Additional details on our implementation can be found in Appendix C and our code is available in supplemental work, and a demonstration of FONDUE is available in a notebook at `https://anonymous.4open.science/r/fondue-demo-CD5A`.

## 5 Results

In this section, we will analyse the results of the experiments detailed in Section 4. First, we will review the IDE of the different datasets in Section 5.1 obtained from MLE and TwoNN, the two ID estimators discussed in Section 2.1. Then, in Section 5.2, we will evaluate the results of FONDUE by comparing the selected number of dimensions with the reconstruction loss and downstream task accuracy.

---

**Algorithm 3** FONDUE$_{IB}$

1: **procedure** FONDUE$_{IB}(t, IDE_{data}, e)$
2:     $l \leftarrow 0$
3:     $u \leftarrow \infty$
4:     $n \leftarrow IDE_{data}$
5:     $m \leftarrow \{\}$
6:     **while** $n \neq l$ **do**
7:         $H(\mathbf{z}), I(\mathbf{x}, \hat{\mathbf{x}}) \leftarrow$ GET-MEM$_{IB}(m, n, e)$
8:         **if** $H(\mathbf{z}) - I(\mathbf{x}, \hat{\mathbf{x}}) \leqslant t$ **then**
9:             $l \leftarrow n$
10:             $n \leftarrow \min(n \times 2, u)$
11:         **else**
12:             $u \leftarrow n$
13:             $n \leftarrow$ floor $\left(\frac{l+u}{2}\right)$
14:         **end if**
15:     **end while**
16:     **return** $n$
17: **end procedure**

**Algorithm 4** GET-MEM$_{IB}$

1: **procedure** GET-MEM$_{IB}(m, n, e)$
2:     **if** $m[n] = \emptyset$ **then**
3:         $vae \leftarrow$ TRAIN-VAE$(dim = n, n\_epochs = e)$
4:         $H(\mathbf{z}), I(\mathbf{x}, \hat{\mathbf{x}}) \leftarrow IB(vae)$
5:         $m[n] \leftarrow H(\mathbf{z}), I(\mathbf{x}, \hat{\mathbf{x}})$
6:     **end if**
7:     **return** $m[n]$
8: **end procedure**

---

### 5.1 Estimating the intrinsic dimensions of the datasets

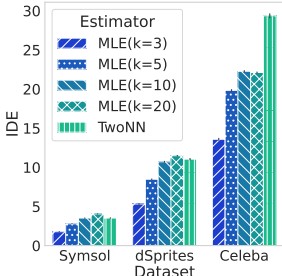

Figure 5: IDEs of dSprites, Celeba, and Symsol using different ID estimation methods.

As mentioned in Section 4, we have selected 3 datasets of increasing intrinsic dimensionality: Symsol (Murphy et al., 2021), dSprites (Higgins et al., 2017), and Celeba (Liu et al., 2015). Following Karbauskaitė et al. (2011), we will retain for our analysis the MLE estimates which are stable for the largest number of $k$ values. We can see in Figure 5 that the MLE estimations become stable when $k$ is between 10 and 20, similar to

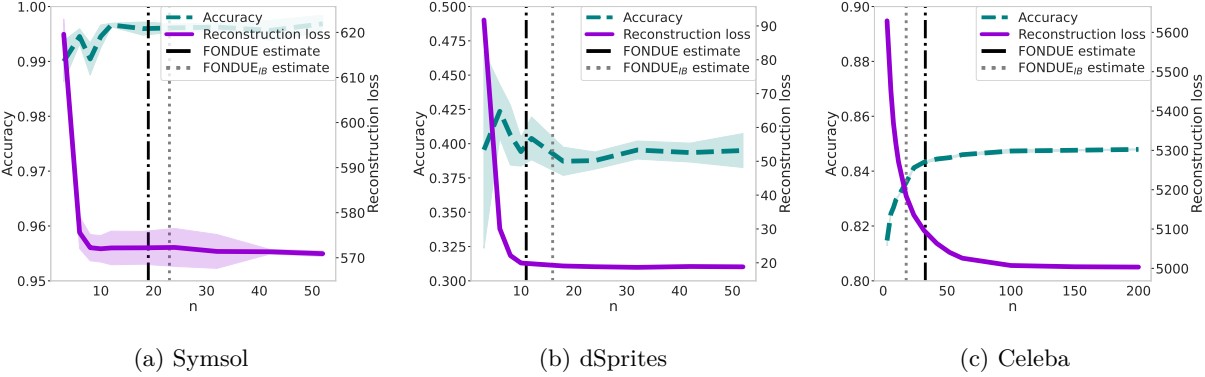

Figure 6: Reconstruction loss and accuracy obtained for generation and downstream tasks of **VAEs** for Symsol, dSprites, and Celeba with an increasing number of latent variables. The black and grey vertical lines indicate the number of dimensions found by FONDUE and FONDUE$_{IB}$.

what was reported by Levina & Bickel (2004). These IDEs are also generally close to TwoNN estimations, except for Celeba, where TwoNN seems to overestimate the ID, as reported by Pope et al. (2021).

Celeba's IDE was previously estimated to be 26 for MLE with $k = 20$ (Pope et al., 2021) neighbours, and we know that Symsol and dSprites have 2 and 5 generative factors, respectively. We thus expect their IDEs to be close to these values. We can see in Figure 5 that MLE and TwoNN overestimate the IDs of Symsol and dSprites, with IDEs of 4 and 11 instead of the expected 2 and 5. Our result for Celeba is close to Pope et al. (2021) with an estimate of 22; the slight difference may be attributed to the difference in our averaging process (Pope et al. (2021) used the averaging described by MacKay & Ghahramani (2005) instead of the original averaging of Levina & Bickel (2004)).

Because stability is more important than overestimation for our purposes, in the rest of this paper, we will consider the IDEs obtained from MLE with $k = 20$. Specifically, we will initialise FONDUE with $n = 4$ for Symsol, $n = 11$ for dSprites and $n = 22$ for Celeba.

## 5.2 Evaluating FONDUE

Using the IDEs reported in the previous section to initialise FONDUE, we will now report the results of FONDUE and FONDUE$_{IB}$ for the considered datasets.

**Obtaining stable estimates** To ensure stable estimates, we computed FONDUE multiple times, gradually increasing the number of epochs $e$ until the predicted dimensionality $n$ stopped changing (i.e., until $\gamma$ is sufficently small). As reported in Table 1, the results were already stable after two epochs[2]. We set a fixed threshold of $t = 1$ in all our experiments and used memoisation (see Algorithm 2) to avoid unnecessary retraining and speed up Algorithm 1. For FONDUE$_{IB}$ we followed the same process, with a fixed threshold of $t = -0.01$.

**Analysing the results of FONDUE** As shown in Table 1, the execution time for finding the number of dimensions for one dataset is much shorter than for fully training one model, which is approximately 2h using the same GPUs. Moreover, on dSprites and Celeba, FONDUE finds the number of dimensions corresponding to low reconstruction loss and good accuracy, which is consistent with the results obtained manually from the Elbow methods (Doersch, 2016; Mai Ngoc & Hwang, 2020). It is worth noting that the number of dimensions provided by FONDUE for dSprites is also close to the dimensionality of 10 generally used in the literature (Higgins et al., 2017; Burgess et al., 2018; Kim & Mnih, 2018; Locatello et al., 2019b). When compared to FONDUE, FONDUE$_{IB}$ is always slower as it requires more epochs to provide stable

---

[2]Note that the numbers of epochs given in Table 1 correspond to the minimum number of epochs needed for FONDUE to be stable. For example, if we obtain the same score after 1 and 2 epochs, the number of epochs given in Table 1 is 1.

Table 1: Number of latent variables $n$ obtained with FONDUE and FONDUE$_{IB}$. The results are averaged over 10 seeds, and computation times are reported for NVIDIA A100 GPUs. The computation time is given for one run of the algorithm over the minimum number of epochs needed to obtain a stable score.

| | Dataset | $n$ (avg $\pm$ SD) | Time/run | Models trained | Epochs/training |
|---|---|---|---|---|---|
| FONDUE | Symsol | $19.1 \pm 0.7$ | 6 min | 8 | 1 |
| FONDUE | dSprites | $10.9 \pm 0.7$ | 42 min | 4 | 2 |
| FONDUE | Celeba | $32.6 \pm 0.7$ | 17 min | 6 | 2 |
| FONDUE$_{IB}$ | Symsol | $23.3 \pm 0.5$ | 40 min | 8 | 6 |
| FONDUE$_{IB}$ | dSprites | $15.7 \pm 0.5$ | 52 min | 5 | 2 |
| FONDUE$_{IB}$ | Celeba | $18.4 \pm 0.7$ | 21 min | 5 | 3 |

estimates. Furthermore, we can see in Figure 6 that it displays an inconsistent behaviour, overestimating the number of dimensions for dSprites and underestimating them for Celeba. Both FONDUE and FONDUE$_{IB}$ overestimate the number of dimensions for Symsol, with a stronger overestimation of FONDUE$_{IB}$. Indeed, it is apparent in Figure 6a that 15 dimensions would be enough to ensure a low reconstruction error and a high accuracy, but FONDUE overestimates it by 4 dimensions and this overestimation is doubled by FONDUE$_{IB}$. We hypothesise that the performance of both FONDUE versions may be impacted by noisy environments as the VAEs trained on Symsol present the largest variance of reconstruction loss. Despite this, Figure 6 shows that FONDUE estimates generally locate the Elbow point correctly, ignoring diminishing returns for more complex datasets like Figure 6c. It thus consistently provides a number of dimensions corresponding to a good trade-off between low reconstruction loss and high accuracy, which is not always the case with FONDUE$_{IB}$.

**Are the results obtained with FONDUE on VAEs applicable to deterministic AEs?** We can see in Figure 7 that FONDUE estimates obtained on VAEs also agree with the Elbow method for deterministic AEs with equivalent architectures. Overall, these results indicate that the dimensionality selected by FONDUE on VAEs can be reused for AEs trained on the same dataset with an identical architecture.

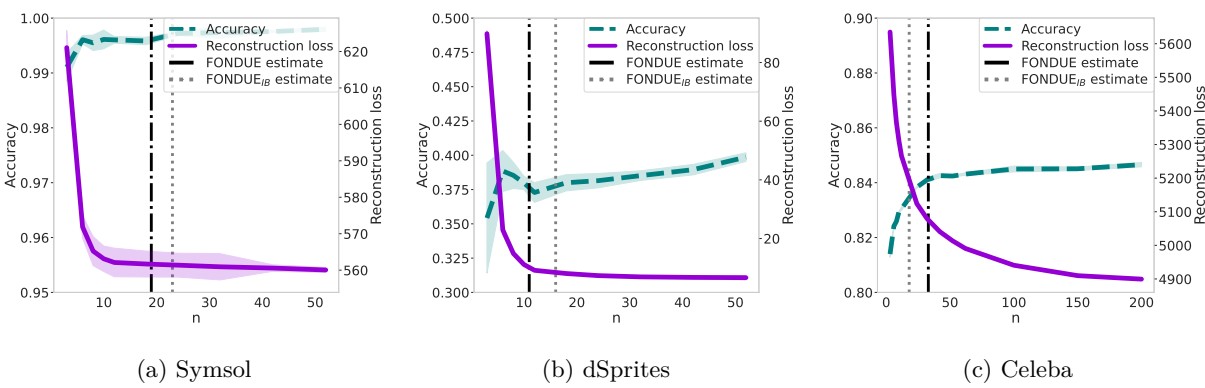

(a) Symsol           (b) dSprites           (c) Celeba

Figure 7: Reconstruction loss and accuracy obtained for generation and downstream tasks of **deterministic AEs** for Symsol, dSprites, and Celeba with an increasing number of latent variables. The black and grey vertical lines indicate the number of dimensions found by FONDUE and FONDUE$_{IB}$ for VAEs with the same architectures.

**Can FONDUE be applied to other architectures, hyperparameters and learning objectives?** To assess the generalisability of FONDUE to other hyperparameters and learning objectives, we compared the results obtained by FONDUE with the Elbow methods for $\beta$-VAEs with $\beta = [0.5, 2, 4]$ and for DIP-VAE II (Kumar et al., 2018), with hyperparameter values $\lambda_{od} = [0.5, 1, 2, 4]$. A full presentation of DIP-VAE II is available in Appendix I. We can see in Figure 8 that FONDUE generalises well to different hyperparameter

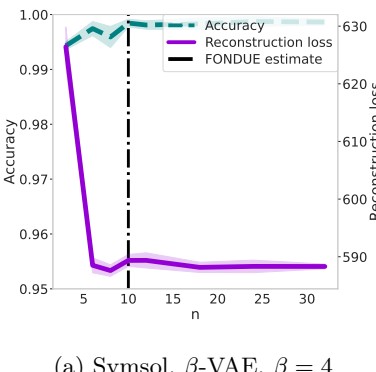
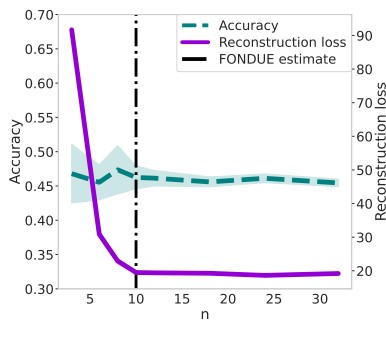

(a) Symsol, $\beta$-VAE, $\beta = 4$         (b) dSprites, DIP-VAE II, $\lambda_{od} = 0.5$

Figure 8: Reconstruction loss and accuracy obtained for generation and downstream tasks with an increasing number of latent variables. (a) shows the results for $\beta$-VAE with $\beta = 4$ on Symsol, and (b) shows the results for DIP-VAE II with $\lambda_{od} = 0.5$ on dSprites. The vertical lines indicate the number of dimensions found by FONDUE.

Table 2: Number of latent variables $n$ obtained with FONDUE with different $\beta$ values for $\beta$-VAE. The results are averaged over 10 seeds, and computation times are reported for NVIDIA A100 GPUs. The computation time is given for one run of the algorithm over the minimum number of epochs needed to obtain a stable score.

| $\beta$ | Dataset | $n$ (avg $\pm$ SD) | Time/run | Models trained | Epochs/training |
|---|---|---|---|---|---|
| 0.5 | Symsol | $26 \pm 0.7$ | 18 min | 8 | 3 |
| 0.5 | dSprites | $13.6 \pm 1.2$ | 80 min | 6 | 3 |
| 0.5 | Celeba | $49.9 \pm 1.5$ | 46 min | 8 | 4 |
| 2 | Symsol | $13.6 \pm 0.8$ | 4 min | 6 | 1 |
| 2 | dSprites | $7.4 \pm 0.5$ | 26 min | 5 | 1 |
| 2 | Celeba | $21.9 \pm 0.3$ | 8 min | 6 | 1 |
| 4 | Symsol | $10 \pm 0.5$ | 4 min | 6 | 1 |
| 4 | dSprites | $6.3 \pm 0.7$ | 24 min | 5 | 1 |
| 4 | Celeba | $15.9 \pm 1.0$ | 23 min | 6 | 3 |

values, and learning objectives, with results on par with the Elbow method, as before. As shown in Tables 2 and 3, the number of dimensions predicted are consistent with the pressure applied on the bottleneck: models with higher $\beta$ (or $\lambda_{od}$) apply a more agressive pruning and have a lower number of active variables than models with lower $\beta$ (or $\lambda_{od}$). Overall, the results obtained are still faster to compute than fully training one model. Additional results for different learning objectives, hyperparameters and datasets can be found in Appendix L. FONDUE also seems to be robust to architectural changes as reported in Appendix D.

## 6   Conclusion

By studying the effect of the polarised regime on the mean and sampled representations, we have shown that one can detect the types of variables (active, passive or mixed) learned by VAEs. These observations lead to FONDUE: an algorithm which can find the number of latent dimensions after which the mean and sampled representations start to strongly diverge. Increasing the number of dimensions beyond this number will result in adding passive or mixed variables which will generally not contribute much to the reconstruction quality and downstream task accuracy. Hence, FONDUE will locate the Elbow point of the performance curves. After proving the correctness of our algorithm, we have shown that it is a faster, automated and unsupervised alternative to existing methods which does not require to fully train any model, is not impacted by architectural changes, and can be used for deterministic AEs.

Table 3: Number of latent variables $n$ obtained with FONDUE with different $\lambda_{od}$ values for DIP-VAE II. The results are averaged over 10 seeds, and computation times are reported for NVIDIA A100 GPUs. The computation time is given for one run of the algorithm over the minimum number of epochs needed to obtain a stable score.

| $\lambda_{od}$ | Dataset | $n$ (avg $\pm$ SD) | Time/run | Models trained | Epochs/training |
|---|---|---|---|---|---|
| 0.5 | Symsol | $17.5 \pm 1.1$ | 7 min | 8 | 1 |
| 0.5 | dSprites | $10.2 \pm 0.6$ | 19 min | 4 | 1 |
| 0.5 | Celeba | $27.3 \pm 1.7$ | 8 min | 6 | 1 |
| 1 | Symsol | $15.9 \pm 2.0$ | 7 min | 7 | 1 |
| 1 | dSprites | $9.5 \pm 1.0$ | 25 min | 5 | 1 |
| 1 | Celeba | $22.7 \pm 2.3$ | 8 min | 6 | 1 |
| 2 | Symsol | $12.4 \pm 1.1$ | 7 min | 6 | 1 |
| 2 | dSprites | $7.9 \pm 0.7$ | 55 min | 6 | 2 |
| 2 | Celeba | $20.9 \pm 1.8$ | 16 min | 6 | 2 |
| 4 | Symsol | $11.7 \pm 0.9$ | 13 min | 6 | 2 |
| 4 | dSprites | $8.1 \pm 0.7$ | 28 min | 6 | 1 |
| 4 | Celeba | $18.7 \pm 1.4$ | 23 min | 6 | 3 |

**Future work**   While FONDUE has been demonstrated to be an efficient algorithm, it could be improved and extended in several ways: 1) we have shown that the dimensions given by FONDUE could also be used for deterministic AEs, but it would be interesting to see if this applies to a larger range of unsupervised models (e.g., GANs, clustering methods, etc.); 2) FONDUE can be extended in a number of ways by replacing the difference of Trace in Algorithm 2 by any function that reliably provides different results in mean and sampled representations early in the training process, as illustrated with IDEs in Appendix F. These extensions could be beneficial both in terms of execution time (if the function is faster or convergence is reached earlier) and complementary theoretical insights (if the function is also theoretically grounded); 3) One could also extend FONDUE to find a good hyperparameter value for $\beta$-VAE by replacing $n$ by $\beta$ in Algorithms 1 and 2.

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

# A Proofs

## A.1 Proof of Proposition 1

*Proof.* We know from Definition 1 that when a latent representation is only composed of active variables, $\lim_{\gamma \to 0} \boldsymbol{Z} = \boldsymbol{M}$. Thus, $\lim_{\gamma \to 0} \text{Var}[\boldsymbol{Z}] = \text{Var}[\boldsymbol{M}]$ and $\lim_{\gamma \to 0} \text{Tr}(\text{Var}[\boldsymbol{Z}]) = \text{Tr}(\text{Var}[\boldsymbol{M}])$. □

## A.2 Proof of Proposition 2

*Proof.* Let us consider the partitioned matrices

$$\text{Var}[\boldsymbol{Z}] = \left[ \begin{array}{c|c} \text{Var}[\boldsymbol{Z_a}] & \text{Cov}[\boldsymbol{Z_a}, \boldsymbol{Z_p}] \\ \hline \text{Cov}[\boldsymbol{Z_p}, \boldsymbol{Z_a}] & \text{Var}[\boldsymbol{Z_p}] \end{array} \right], \tag{3}$$

and

$$\text{Var}[\boldsymbol{M}] = \left[ \begin{array}{c|c} \text{Var}[\boldsymbol{M_a}] & \text{Cov}[\boldsymbol{M_a}, \boldsymbol{M_p}] \\ \hline \text{Cov}[\boldsymbol{M_p}, \boldsymbol{M_a}] & \text{Var}[\boldsymbol{M_p}] \end{array} \right], \tag{4}$$

where $\cdot_{\boldsymbol{a}}$ and $\cdot_{\boldsymbol{p}}$ denote the subsets of active and passive variables, respectively. We know from Definition 1 that $\lim_{\gamma \to 0} \text{Var}[\boldsymbol{Z_p}] = \boldsymbol{I}_{s \times s}$ and $\lim_{\gamma \to 0} \text{Var}[\boldsymbol{M_p}] = \boldsymbol{0}_{s \times s}$. Thus,

$$\lim_{\gamma \to 0} \text{Tr}(\text{Var}[\boldsymbol{Z}]) = \text{Tr}(\text{Var}[\boldsymbol{Z_a}]) + \text{Tr}(\text{Var}[\boldsymbol{Z_p}]), \tag{5}$$

$$= \text{Tr}(\text{Var}[\boldsymbol{Z_a}]) + \text{Tr}(\boldsymbol{I}_{s \times s}), \tag{6}$$

$$= \text{Tr}(\text{Var}[\boldsymbol{Z_a}]) + s, \tag{7}$$

and

$$\lim_{\gamma \to 0} \text{Tr}(\text{Var}[\boldsymbol{M}]) = \text{Tr}(\text{Var}[\boldsymbol{M_a}]) + \text{Tr}(\text{Var}[\boldsymbol{M_p}]), \tag{8}$$

$$= \text{Tr}(\text{Var}[\boldsymbol{M_a}]) + \text{Tr}(\boldsymbol{0}_{s \times s}), \tag{9}$$

$$= \text{Tr}(\text{Var}[\boldsymbol{Z_a}]). \tag{10}$$

Combining equations 7 and 10 and recalling from Proposition 1 that $\lim_{\gamma \to 0} \text{Tr}(\text{Var}[\boldsymbol{Z_a}]) = \text{Tr}(\text{Var}[\boldsymbol{M_a}])$, we obtain

$$\lim_{\gamma \to 0} \Big( \text{Tr}(\text{Var}[\boldsymbol{Z}]) - \text{Tr}(\text{Var}[\boldsymbol{M}]) \Big) = \text{Tr}(\text{Var}[\boldsymbol{Z_a}]) + s - \text{Tr}(\text{Var}[\boldsymbol{Z_a}]) = s, \tag{11}$$

as expected. □

## A.3 Proof of Proposition 3

*Proof.* Let us consider the partitioned matrices

$$\text{Var}[\boldsymbol{Z}] = \left[ \begin{array}{c|c} \text{Var}[\boldsymbol{Z_a}] & \text{Cov}[\boldsymbol{Z_a}, \boldsymbol{Z_m}] \\ \hline \text{Cov}[\boldsymbol{Z_m}, \boldsymbol{Z_a}] & \text{Var}[\boldsymbol{Z_m}] \end{array} \right], \tag{12}$$

and

$$\text{Var}[\boldsymbol{\mu}] = \left[ \begin{array}{c|c} \text{Var}[\boldsymbol{M_a}] & \text{Cov}[\boldsymbol{M_a}, \boldsymbol{M_m}] \\ \hline \text{Cov}[\boldsymbol{M_m}, \boldsymbol{M_a}] & \text{Var}[\boldsymbol{M_m}] \end{array} \right], \tag{13}$$

where $\cdot_{\boldsymbol{a}}$ and $\cdot_{\boldsymbol{m}}$ denote the subsets of active and mixed variables, respectively. We know from Definition 1 that up to some permutations a mixed variable $\boldsymbol{Z}_i = \left[ \boldsymbol{Z}_i^{(\boldsymbol{a})}, \boldsymbol{Z}_i^{(\boldsymbol{p})} \right]$, where $\boldsymbol{a}$ is the subset of data examples indexes for which the variable is active and $\boldsymbol{p}$ the subset of data example indexes for which the variable is passive. Given $h$ data examples, we thus have $h = \text{card}(\boldsymbol{a}) + \text{card}(\boldsymbol{b})$ where $\text{card}(\cdot)$ denotes the cardinality of a set. Let us define $w_i \triangleq \frac{\text{card}(\boldsymbol{a})}{h}$ with $0 < w_i < 1$. We have $(1 - w_i) = \frac{h - \text{card}(\boldsymbol{a})}{h} = \text{card}(\boldsymbol{b})$. Given the mean of the mixed variable $i$ of the sampled representation $\bar{\boldsymbol{Z}}_i$, we thus obtain:

$$\lim_{\gamma \to 0} \bar{\boldsymbol{Z}}_i = w_i \bar{\boldsymbol{Z}}_i^{(\boldsymbol{a})} + (1 - w_i) \bar{\boldsymbol{Z}}_i^{(\boldsymbol{p})} = w_i \bar{\boldsymbol{Z}}_i^{(\boldsymbol{a})}. \tag{14}$$

Now, let us calculate the variance of $\boldsymbol{Z}_i$, $\mathrm{Var}[\boldsymbol{Z}_i]$:

$$\lim_{\gamma \to 0} \mathrm{Var}[\boldsymbol{Z}_i] = w_i \left( \bar{\boldsymbol{Z}}_i^{(\boldsymbol{a})} - \bar{\boldsymbol{Z}}_i \right)^2 + (1 - w_i) w_i \left( \bar{\boldsymbol{Z}}_i^{(\boldsymbol{p})} - \bar{\boldsymbol{Z}}_i \right)^2 + w_i \mathrm{Var}\left[ \boldsymbol{Z}_i^{(\boldsymbol{a})} \right] + (1 - w_i) \mathrm{Var}\left[ \boldsymbol{Z}_i^{(\boldsymbol{p})} \right], \quad (15)$$

$$= w_i (1 - w_i)^2 \left( \bar{\boldsymbol{Z}}_i^{(\boldsymbol{a})} \right)^2 + w_i^2 (1 - w_i) \left( \bar{\boldsymbol{Z}}_i^{(\boldsymbol{a})} \right)^2 + w_i \mathrm{Var}\left[ \boldsymbol{Z}_i^{(\boldsymbol{a})} \right] + 1 - w_i, \quad (16)$$

$$= w_i (1 - w_i) \left( \bar{\boldsymbol{Z}}_i^{(\boldsymbol{a})} \right)^2 + w_i \mathrm{Var}\left[ \boldsymbol{Z}_i^{(\boldsymbol{a})} \right] + 1 - w_i, \quad (17)$$

$$= \kappa_i + 1 - w_i, \quad (18)$$

where $\kappa_i = w_i (1 - w_i) \left( \bar{\boldsymbol{Z}}_i^{(\boldsymbol{a})} \right)^2 + w_i \mathrm{Var}\left[ \boldsymbol{Z}_i^{(\boldsymbol{a})} \right]$. Writing $\boldsymbol{M}_i = \left[ \boldsymbol{M}_i^{(\boldsymbol{a})}, \boldsymbol{M}_i^{(\boldsymbol{p})} \right]$, we obtain in the same way

$$\lim_{\gamma \to 0} \bar{\boldsymbol{M}}_i = w_i \bar{\boldsymbol{M}}_i^{(\boldsymbol{a})} + (1 - w_i) \bar{\boldsymbol{M}}_i^{(\boldsymbol{p})} = w_i \bar{\boldsymbol{M}}_i^{(\boldsymbol{a})} = w_i \bar{\boldsymbol{Z}}_i^{(\boldsymbol{a})}, \quad (19)$$

and

$$\lim_{\gamma \to 0} \mathrm{Var}[\boldsymbol{M}_i] = w_i \left( \bar{\boldsymbol{M}}_i^{(\boldsymbol{a})} - \bar{\boldsymbol{M}}_i \right)^2 + (1 - w_i) w_i \left( \bar{\boldsymbol{M}}_i^{(\boldsymbol{p})} - \bar{\boldsymbol{M}}_i \right)^2 + w_i \mathrm{Var}\left[ \boldsymbol{M}_i^{(\boldsymbol{a})} \right] \quad (20)$$

$$+ (1 - w_i) \mathrm{Var}\left[ \boldsymbol{M}_i^{(\boldsymbol{p})} \right], \quad (21)$$

$$= w_i (1 - w_i) \left( \bar{\boldsymbol{M}}_i^{(\boldsymbol{a})} \right)^2 + w_i \mathrm{Var}\left[ \boldsymbol{M}_i^{(\boldsymbol{a})} \right], \quad (22)$$

$$= w_i (1 - w_i) \left( \bar{\boldsymbol{Z}}_i^{(\boldsymbol{a})} \right)^2 + w_i \mathrm{Var}\left[ \boldsymbol{Z}_i^{(\boldsymbol{a})} \right], \quad (23)$$

$$= \kappa_i. \quad (24)$$

Using Equation 18, we have

$$\lim_{\gamma \to 0} \mathrm{Tr}(\mathrm{Var}[\boldsymbol{Z}]) = \mathrm{Tr}(\mathrm{Var}[\boldsymbol{Z_a}]) + \mathrm{Tr}(\mathrm{Var}[\boldsymbol{Z_m}]) = \mathrm{Tr}(\mathrm{Var}[\boldsymbol{Z_a}]) + \sum_{i=1}^{s} (\kappa_i + 1 - w_i). \quad (25)$$

In the same way, from Equation 24 we obtain

$$\lim_{\gamma \to 0} \mathrm{Tr}(\mathrm{Var}[\boldsymbol{M}]) = \mathrm{Tr}(\mathrm{Var}[\boldsymbol{M_a}]) + \mathrm{Tr}(\mathrm{Var}[\boldsymbol{M_m}]) = \mathrm{Tr}(\mathrm{Var}[\boldsymbol{Z_a}]) + \sum_{i=1}^{s} \kappa_i. \quad (26)$$

Thus, $\lim_{\gamma \to 0} \mathrm{Tr}(\mathrm{Var}[\boldsymbol{Z}]) - \mathrm{Tr}(\mathrm{Var}[\boldsymbol{M}]) = \sum_{i=1}^{s} (1 - w_i)$. Given that for all $i$, $0 < w_i < 1$, for the $s$ mixed variables $0 < \sum_{i=1}^{s} (1 - w_i) < s$, and $0 < \lim_{\gamma \to 0} \mathrm{Tr}(\mathrm{Var}[\boldsymbol{Z}]) - \mathrm{Tr}(\mathrm{Var}[\boldsymbol{M}]) < s$, as required. $\qquad \square$

### A.4 Proof of Theorem 1

*Proof.* We know from Proposition 1 that if all the variables are active, $\lim_{\gamma \to 0} \mathrm{Tr}(\mathrm{Var}[\boldsymbol{Z}]) - \mathrm{Tr}(\mathrm{Var}[\boldsymbol{M}]) = 0$. Once their maximum number $n$ is reached, the next variable learned will be either passive or mixed. Recall from Proposition 2 that for $s = 1$ passive variables, $\lim_{\gamma \to 0} \mathrm{Tr}(\mathrm{Var}[\boldsymbol{Z}]) - \mathrm{Tr}(\mathrm{Var}[\boldsymbol{M}]) = 1$. Moreover, using Proposition 3 with $s = 1$ mixed variables, $0 < \lim_{\gamma \to 0} \mathrm{Tr}(\mathrm{Var}[\boldsymbol{Z}]) - \mathrm{Tr}(\mathrm{Var}[\boldsymbol{M}]) < 1$. As a result, if $n$ is the maximum number of active variables, at $n + 1$, $0 < \lim_{\gamma \to 0} \mathrm{Tr}(\mathrm{Var}[\boldsymbol{Z}]) - \mathrm{Tr}(\mathrm{Var}[\boldsymbol{M}]) \leqslant 1$, where the upper bound is tight when the $(n+1)^{th}$ variable is passive. $\qquad \square$

### A.5 Proof of Theorem 2

*Proof.* Here we consider the case where $n$ is the maximum number of mixed and active variables that can be reached. Using Proposition 3 with $0 \leqslant s \leqslant n$, at $n$ we have:

$$0 \leqslant \lim_{\gamma \to 0} \mathrm{Tr}(\mathrm{Var}[\boldsymbol{Z}]) - \mathrm{Tr}(\mathrm{Var}[\boldsymbol{M}]) < n, \quad (27)$$

where the lower bound is tight when $s = 0$. As the $(n+1)^{th}$ variable will always be passive, using Proposition 2 with $s = 1$, we obtain

$$1 \leqslant \lim_{\gamma \to 0} \text{Tr}(\text{Var}[\boldsymbol{Z}]) - \text{Tr}(\text{Var}[\boldsymbol{M}]) < n + 1, \tag{28}$$

as expected. $\qquad\square$

### A.6 Proof of Theorem 3

This section provides the full proof of Theorem 3. To ease its reading, let us first recall from Theorems 1 and 2 that the mean and sampled representations start to diverge only after the number of latent dimensions has become large enough for non-active variables to appear.

**Remark 1.** *Given that $l$ and $u$ only take values of latent dimensions for which $\text{Tr}(\text{Var}[\boldsymbol{Z}]) - \text{Tr}(\text{Var}[\boldsymbol{M}]) \leqslant t$ and $\text{Tr}(\text{Var}[\boldsymbol{Z}]) - \text{Tr}(\text{Var}[\boldsymbol{M}]) > t$, respectively, Theorems 1 and 2 imply that for all iterations $i$, $l_i < u_i$.*

Using the loop invariant $l_i \leqslant n_i \leqslant u_i$ for each iteration $i$, we will now show that Algorithm 1 terminates when $l_i = n_i = \text{floor}\left(\frac{l_i + u_i}{2}\right)$, which can only be reached when $u_i = n_i + 1$, that is when $n_i$ is the maximum number of latent dimensions for which we have $\text{Tr}(\text{Var}[\boldsymbol{Z}]) - \text{Tr}(\text{Var}[\boldsymbol{M}]) \leqslant t$.

*Proof.*

**Initialisation:** $l_0 = 0, n_0 = IDE_{data}, u_0 = \infty$, thus $l_0 < n_0 < u_0$.

**Maintenance:** We will consider both branches of the if statement separately:

- For $\text{Tr}(\text{Var}[\boldsymbol{Z}]) - \text{Tr}(\text{Var}[\boldsymbol{M}]) \leqslant t$ (lines 9-11), $u_i = u_{i-1}$, $n_i = min(n_{i-1} \times 2, u_i)$, and $l_i = n_{i-1}$. We directly see that $n_i \leqslant u_i$. We know from Remark 1 that $l_i < u_i$ and we also have $l_i < n_{i-1} \times 2$, it follows that $l_i < n_i$. Grouping both inequalities, we get $l_i < n_i \leqslant u_i$.

- For $\text{Tr}(\text{Var}[\boldsymbol{Z}]) - \text{Tr}(\text{Var}[\boldsymbol{M}]) > t$ (lines 12-14), $u_i = p_{i-1}, l_i = l_{i-1}$, and $n_i = \text{floor}\left(\frac{l_i + u_i}{2}\right)$. Using Remark 1 we can directly see that $l_i \leqslant \text{floor}\left(\frac{l_i + u_i}{2}\right) < u_i$ and we obtain $l_i \leqslant n_i < u_i$.

**Termination:** The loop terminates when $l_i = n_i$. Given that $l_i < n_i$ when $\text{Tr}(\text{Var}[\boldsymbol{Z}]) - \text{Tr}(\text{Var}[\boldsymbol{M}]) \leqslant t$, this is only possible when $\text{Tr}(\text{Var}[\boldsymbol{Z}]) - \text{Tr}(\text{Var}[\boldsymbol{M}]) > t$, which is when $n_i = \text{floor}\left(\frac{l_i + u_i}{2}\right)$. We know from Remark 1 that $l_i < u_i$, so we must have $(l_i + u_i) \bmod 2 > 0$. As $a \bmod 2 \in \{0, 1\}$, the only possible value for $u_i$ to satisfy $l_i = n_i = \text{floor}\left(\frac{l_i + u_i}{2}\right)$ is $u_i = n_i + 1$. Thus, $n_i$ is the largest number of latent dimensions for which $\text{Tr}(\text{Var}[\boldsymbol{Z}]) - \text{Tr}(\text{Var}[\boldsymbol{M}]) \leqslant t$. $\qquad\square$

## B Resources

As mentioned in Sections 1 and 4, we released the code of our experiment, a demo of FONDUE and the pre-trained models:

- The demonstration of FONDUE is available at `https://anonymous.4open.science/r/fondue-demo-CD5A`

- `symsol_reduced`, the reduced version of Symmetric solids, can be downloaded using an anonymous Google account using the following tiny URL `https://t.ly/G7Qp`

- The code can be found in supplemental work

- Our pre-trained models are large and could not be shared with the reviewers using an anonymous link. The URL to the models will, however, be available in the non-anonymised version of this paper.

The 300 models correspond to 5 runs of VAEs trained with:

- 8 choices of latent dimensions for Symsol and dSprites, using convolutional and fully-connected architectures, resulting in 160 models

- 14 choices of latent dimensions for Celeba, using convolutional and fully-connected architectures, resulting in 140 models

The total 300 pre-trained models were then used to compute estimate IDs as described below. Note that while these models would save some computational time if used to reproduce the experiment, they are only provided to reduce the carbon footprint of reproducing the experiment as one could easily retrain the models using the details of our implementation.

## C  Experimental setup

Our implementation uses the same hyperparameters as Locatello et al. (2019b), as listed in Table 4. We reimplemented the Locatello et al. (2019b) code base, designed for Tensorflow 1, in Tensorflow 2 using Keras. The model architectures used are also similar, as described in Tables 5 and 6. We used the convolutional architecture in the main paper and the fully-connected architecture in Appendix D. Each model is trained 5 times with seed values from 0 to 4. Every image input is normalised to have pixel values between 0 and 1. TwoNN is used with an anchor of 0.9, and the hyperparameters for MLE can be found in Table 7. More details about these two ID estimators can be found in Section 2.1 and Appendix E.

Table 4: VAEs hyperparameters

| Parameter | Value |
|---|---|
| Batch size | 64 |
| Latent space dimension | 3, 6, 8, 10, 12, 18, 24, 32. |
| | For Celeba only: 42, 52, 62, 100, 150, 200 |
| Optimizer | Adam |
| Adam: $\beta_1$ | 0.9 |
| Adam: $\beta_2$ | 0.999 |
| Adam: $\epsilon$ | 1e-8 |
| Adam: learning rate | 0.0001 |
| Reconstruction loss | Bernoulli |
| Training steps | 300,000 |
| Train/test split | 90/10 |
| $\beta$ | 1 |

Table 5: Architecture

| Encoder | Decoder |
|---|---|
| Input: $\mathbb{R}^{64 \times 63 \times channels}$ | $\mathbb{R}^{10}$ |
| Conv, kernel=4×4, filters=32, activation=ReLU, strides=2 | FC, output shape=256, activation=ReLU |
| Conv, kernel=4×4, filters=32, activation=ReLU, strides=2 | FC, output shape=4x4x64, activation=ReLU |
| Conv, kernel=4×4, filters=64, activation=ReLU, strides=2 | Deconv, kernel=4×4, filters=64, activation=ReLU, strides=2 |
| Conv, kernel=4×4, filters=64, activation=ReLU, strides=2 | Deconv, kernel=4×4, filters=32, activation=ReLU, strides=2 |
| FC, output shape=256, activation=ReLU, strides=2 | Deconv, kernel=4×4, filters=32, activation=ReLU, strides=2 |
| FC, output shape=2x10 | Deconv, kernel=4×4, filters=channels, activation=ReLU, strides=2 |

Table 6: Fully-connected architecture

| Encoder | Decoder |
|---|---|
| Input: $\mathbb{R}^{64 \times 63 \times channels}$ | $\mathbb{R}^{10}$ |
| FC, output shape=1200, activation=ReLU | FC, output shape=256, activation=tanh |
| FC, output shape=1200, activation=ReLU | FC, output shape=1200, activation=tanh |
| FC, output shape=2x10 | FC, output shape=1200, activation=tanh |

Table 7: MLE hyperparameters

| Parameter | Value |
|---|---|
| k | 3, 5, 10, 20 |
| anchor | 0.8 |
| seed | 0 |
| runs | 5 |

## D    FONDUE on fully-connected architectures

We report the results obtained by FONDUE for fully-connected (FC) architectures in Table 8 and Figure 9. As shown in Table 8, the execution time for estimating the number of dimensions for one dataset is much shorter than for training one model (this is approximately 2h on the same GPUs), consistently with convolutional VAEs. As in Section 5.2, FONDUE correctly finds the number of latent dimensions that would be selected by Elbow methods, as shown in Figure 9. $\text{FONDUE}_{IB}$ predictions are more consistent with those of FONDUE for FC architectures, except for dSprites where the number of dimensions is largely overestimated.

As in Section 5.2, we gradually increase the number of epochs until FONDUE reaches a stable estimation of the latent dimensions. As these models have fewer parameters than the convolutional architecture used in Section 5.2, they converge more slowly and need to be trained for more epochs on Celeba and Symsol before reaching a stable estimation (Arora et al., 2018; Sankararaman et al., 2020). As before, $\text{FONDUE}_{IB}$ generally requires more epochs than FONDUE to converge, except for Celeba.

Overall, we can see that FONDUE also provides good results on the FC architectures, despite a slower convergence, showing robustness to architectural changes.

## E    Additional details on ID estimation

In this section, we will detail two ID estimation techniques which use the statistical properties of the neighbourhood of each data point to estimate $d$, and provide good results for approximating the ID of deep neural

Table 8: Number of latent variables $n$ obtained with FONDUE and $\text{FONDUE}_{IB}$ for **fully-connected VAEs**. The results are averaged over 10 seeds, and computation times are reported for NVIDIA A100 GPUs. The computation time is given for one run of the algorithm over the minimum number of epochs needed to obtain a stable score.

| | Dataset | $n$ (avg $\pm$ SD) | Time/run | Models trained | Epochs/training |
|---|---|---|---|---|---|
| FONDUE | Symsol | 7.6 ± 0.7 | 9 min | 5 | 3 |
| FONDUE | dSprites | 5.2 ± 0.6 | 16 min | 5 | 1 |
| FONDUE | Celeba | 22.0 ± 0. | 23 min | 6 | 4 |
| $\text{FONDUE}_{IB}$ | Symsol | 11 ± 0.5 | 14 min | 6 | 4 |
| $\text{FONDUE}_{IB}$ | dSprites | 12.3 ± 0.5 | 28 min | 5 | 2 |
| $\text{FONDUE}_{IB}$ | Celeba | 16.4 ± 0.7 | 5 min | 5 | 1 |

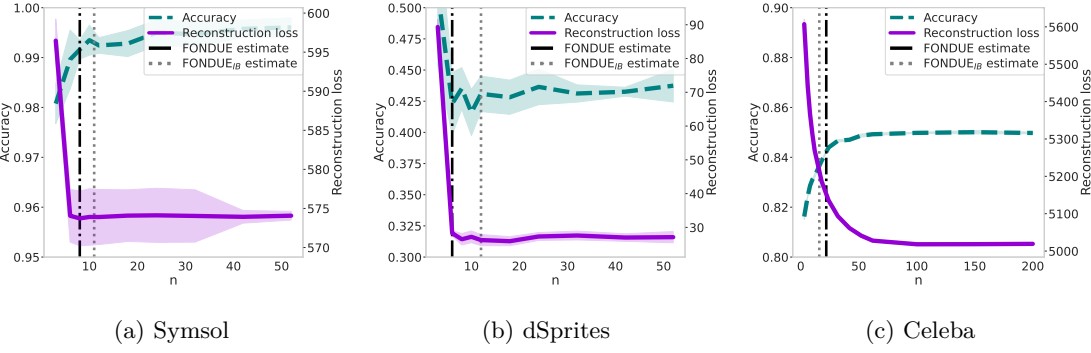

(a) Symsol                    (b) dSprites                    (c) Celeba

Figure 9: Reconstruction loss and accuracy obtained for generation and downstream tasks of **fully-connected VAEs** for test data from Symsol, dSprites, and Celeba with an increasing number of latent variables. The plain and dashed vertical lines indicate the number of dimensions found by FONDUE and $\text{FONDUE}_{IB}$.

network representations and deep learning datasets (Ansuini et al., 2019; Gong et al., 2019; Pope et al., 2021).

### E.1  Maximum Likelihood Estimation

Levina & Bickel (2004) modelled the neighbourhood of a given point $\boldsymbol{X}_i$ as a Poisson process in a d-dimensional sphere $S_{\boldsymbol{X}_i}(R)$ of radius $R$ around $\boldsymbol{X}_i$. This Poisson process is denoted $\{N(t, \boldsymbol{X}_i), 0 \leqslant t \leqslant R\}$, where $N(t, \boldsymbol{X}_i)$ is a random variable representing the number of neighbours of $\boldsymbol{X}_i$ within a radius $t$, and is distributed according to a Poisson distribution[3]. This Poisson process, $\{N(t, \boldsymbol{X}_i), 0 \leqslant t \leqslant R\}$, will count the total number of points falling into the successive $d$-dimensional spheres of radius $0 \leqslant t \leqslant R$.

Intuitively, when $d = 3$ this can be thought of as an onion to which we add an outer peel for each increasing radius value $t$, until we reach the maximum radius $R$. Thus, each $N(t, \boldsymbol{X}_i)$ will give us a snapshot of the number of points contained in all the peels stacked so far in the onion of radius $t$. As $N(t, \boldsymbol{X}_i)$ is a function of the surface area of the sphere, its rate is a function of $d$ and one can estimate $d$ using MLE. However, we generally cannot access all the existing neighbours of $\boldsymbol{X}_i$ in a given radius without infinite data, so we approximate the process using a fixed number of neighbours.

More formally, each point $\boldsymbol{X}_j \in S_{\boldsymbol{X}_i}(R)$ is thus considered as an event, its arrival time $t = T(\boldsymbol{X}_i, \boldsymbol{X}_j)$ being the Euclidean distance from $\boldsymbol{X}_i$ to its $j^{th}$ neighbour $\boldsymbol{X}_j$. By expressing the rate $\lambda(t, \boldsymbol{X}_i)$ of the process $N(t, \boldsymbol{X}_i)$ as a function of the surface area of the sphere—and thus relating $\lambda(t, \boldsymbol{X}_i)$ to $d$—they obtain a maximum likelihood estimation (MLE) of the ID $d$:

$$\bar{d}_R(\boldsymbol{X}_i) = \left[ \frac{1}{N(R, \boldsymbol{X}_i)} \sum_{j=1}^{N(R, \boldsymbol{X}_i)} \log \frac{R}{T(\boldsymbol{X}_i, \boldsymbol{X}_j)} \right]^{-1}. \tag{29}$$

Equation 29 is then simplified by fixing the number of neighbours, $k$, instead of the radius $R$ of the sphere, such that

$$\bar{d}_k(\boldsymbol{X}_i) = \left[ \frac{1}{k-1} \sum_{j=1}^{k-1} \log \frac{T(\boldsymbol{X}_i, \boldsymbol{X}_k)}{T(\boldsymbol{X}_i, \boldsymbol{X}_j)} \right]^{-1}, \tag{30}$$

where the last summand is omitted, as it is zero for $j = k$. The final IDE $\bar{d}_k$ is the averaged score over $n$ data examples (Levina & Bickel, 2004)

$$\bar{d}_k = \frac{1}{n} \sum_{i=1}^{n} \bar{d}_k(\boldsymbol{X}_i). \tag{31}$$

---

[3]Note that this does not imply any distributional assumption about the dataset

To obtain an accurate estimation of the ID with MLE, it is very important to choose a sufficient number of neighbours $k$ to form a dense small sphere (Levina & Bickel, 2004). On one hand, if $k$ is too small, MLE will generally underestimate the ID, and suffer from high variance (Levina & Bickel, 2004; Campadelli et al., 2015; Pope et al., 2021). On the other hand, if $k$ is too large, the ID will be overestimated (Levina & Bickel, 2004; Pope et al., 2021).

**A worked example** Now, let us consider the point $\boldsymbol{X}_i = (0,0,0)$ and 3 closest neighbours $\boldsymbol{Y} = \{(0,1,0)(1,0,0),(2,0,0)\}$. We have $N(t=1,X_i) = 2$ and $N(t=2,X_i) = 3$ because $\boldsymbol{Y}_1, \boldsymbol{Y}_2$ are within a radius $t=1$ of $\boldsymbol{X}_i$, and all $\boldsymbol{Y}_j$ are within a radius $t=2$.

Using the distances between $\boldsymbol{X}_i$ and its neighbours, $T(\boldsymbol{X}_i, \boldsymbol{Y}_j)$, the dimensionality can be estimated by Equation 30 as follows

$$
\begin{aligned}
\bar{d}_3(\boldsymbol{X}_i) &= \left[ \frac{1}{2} \sum_{j=1}^{2} \log \frac{T(\boldsymbol{X}_i, \boldsymbol{Y}_3)}{T(\boldsymbol{X}_i, \boldsymbol{Y}_j)} \right]^{-1}, \\
&= \left[ \frac{1}{2} \sum_{j=1}^{2} \log \frac{2}{T(\boldsymbol{X}_i, \boldsymbol{Y}_j)} \right]^{-1}, \\
&= [\log 2]^{-1}, \\
&\approx 3.3,
\end{aligned}
\tag{32}
$$

which is reasonably close to the true data ID.

To make sure that the estimate is stable, we repeat this estimation over multiple data points and average the results as per Equation 31.

### E.2 TwoNN

Facco et al. (2017) proposed an estimation of the ID based on the ratio of the two nearest neighbours of $\boldsymbol{X}_i$, $r_{\boldsymbol{X}_i} = \frac{T(\boldsymbol{X}_i, \boldsymbol{X}_l)}{T(\boldsymbol{X}_i, \boldsymbol{X}_j)}$, where $\boldsymbol{X}_j$ and $\boldsymbol{X}_l$ are the first and second closest neighbours of $\boldsymbol{X}_i$, respectively. $r$ follows a Pareto distribution with scale $s=1$ and shape $d$, and its density function $f(r)$ is

$$
f(r) = \frac{ds^d}{r^{d+1}} = dr^{-(d+1)}.
\tag{33}
$$

Its cumulative distribution function is thus

$$
F(r) = 1 - \frac{s^d}{r^d} = 1 - r^{-d},
\tag{34}
$$

and, using Equation 34, one can readily obtain $d = \frac{-\log(1-F(r))}{\log r}$. From this, we can see that $d$ is the slope of the straight line passing through the origin, which is given by the set of coordinates $\mathbb{S} = \{(\log r_{\boldsymbol{X}_i}, -\log(1 - F(r_{\boldsymbol{X}_i}))) \mid i = 1, \cdots, m\}$, and can be recovered by linear regression.

As TwoNN uses only two neighbours, it can be sensitive to outliers (Facco et al., 2017) and does not perform well on high ID (Pope et al., 2021), overestimating the ID in both cases.

### E.3 Ensuring an accurate analysis

Given the limitations previously mentioned, we take two remedial actions to guarantee that our analysis is as accurate as possible. To provide an IDE which is as accurate as possible with MLE, we will measure the MLE with an increasing number of neighbours and, similar to Karbauskaitė et al. (2011), retain the IDE which is stable for the largest number of $k$ values. TwoNN will be used as a complementary metric to validate our choice of $k$ for MLE. In case of significant discrepancies with a significantly higher TwoNN IDE, we will rely on the results provided by MLE.

# F   FONDUE based on intrinsic dimension estimation

In this section, we are interested in replacing $\mathrm{Tr}(\mathrm{Var}[\mathbf{z}]) - \mathrm{Tr}(\mathrm{Var}[\boldsymbol{\mu}])$ by $IDE_{\mathbf{z}} - IDE_{\boldsymbol{\mu}}$ in Algorithm 1. While we will not provide a formal proof of the relationship between the polarised regime and ID estimation, we will show that ID estimation match the constraints discussed in Section 3.2 as it is 1) sensitive to the different types of variables in mean and sampled representations 2) stable early during the training. We generally obtain similar results as in Section 5, and any discrepancies may be attributed to the choice of $t$ value as, in opposition to Section 3.2, we lack a principled way of choosing it.

## F.1   IDEs of the mean and sampled representations of VAEs

**Mean and sampled representations have different IDEs**   Looking into the IDEs of mean and sampled representations in Figure 10, we see a clear pattern emerges: when increasing the number of latent variables, the IDEs remain similar up to a point, then abruptly diverge. As discussed in Section 2.4, once a VAE has enough latent variables to encode the information needed by the decoder, the remaining variables will become passive to minimise the KL divergence in Equation 2. This phenomenon will naturally occur when we increase the number of latent variables. We can thus hypothesise that the difference between the mean and sampled IDEs grows with the number of mixed and passive variables. This is verified by computing the number of active, mixed, and passive variables using the method of Bonheme & Grzes (2021), as shown in Figure 11.

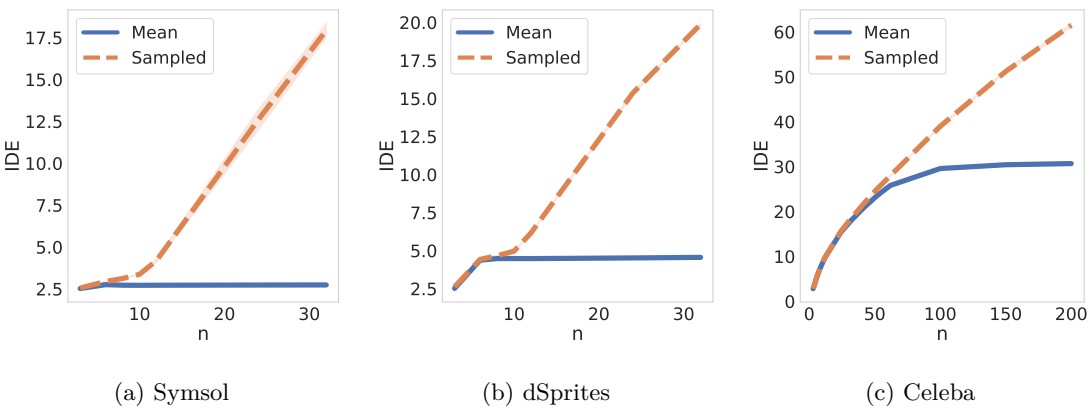

(a) Symsol          (b) dSprites          (c) Celeba

Figure 10: IDE of the mean and sampled representations of VAEs trained with an increasing number of latent dimensions $n$. (a), (b), and (c) shows the results on Symsol, dSprites, and Celeba, respectively.

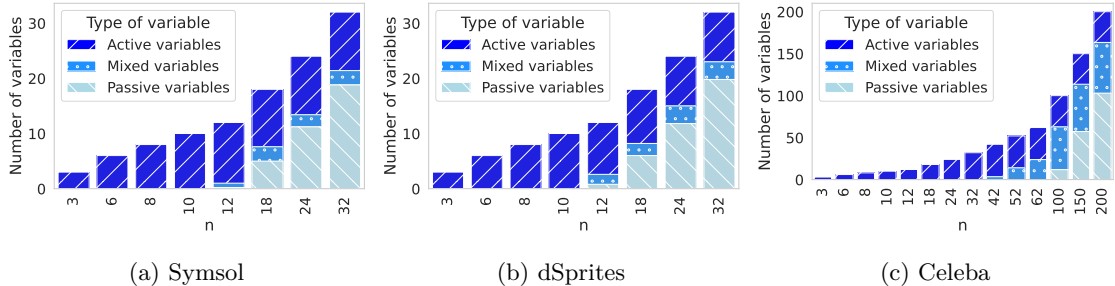

(a) Symsol          (b) dSprites          (c) Celeba

Figure 11: Quantity of active, mixed, and passive variables of VAEs trained with an increasing number of latent dimensions $n$. (a), (b), and (c) show the results on Symsol, dSprites, and Celeba.

**The IDEs of the model's representations do not change much after the first epoch**   The IDEs of the different layers do not change much after the first epoch for well-performing models (see Figure 12).

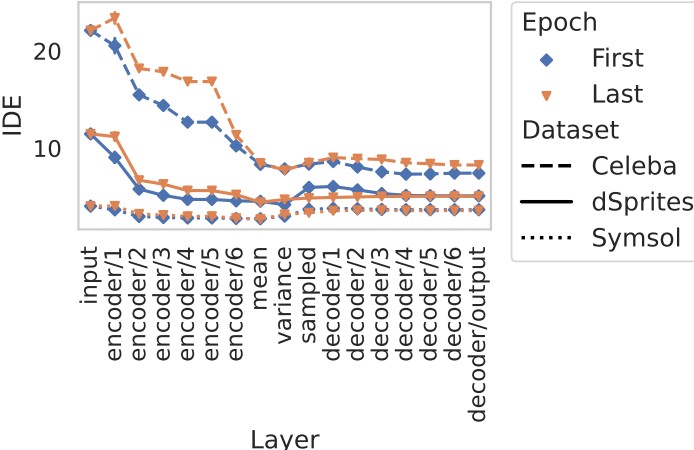

Figure 12: The evolution over multiple epochs of the IDE of the representations learned by VAEs using 10 latent variables on Symsol, dSprites, and Celeba.

However, for Celeba, whose number of latent dimensions is lower than the data IDE and thus cannot reconstruct the data well, the IDEs tend to change more in the early layers of the encoder, displaying a higher variance.

## F.2 FONDUE with IDE

As discussed above, the IDEs of the mean and sampled representations start to diverge when (unused) passive variables appear, and this is already visible after the first epochs of training. The difference of IDEs between the mean and sampled representations thus meet the two criteria for extension listed in Section 3.2 and can be used as a new flavour of FONDUE, $\text{FONDUE}_{IDE}$. As for $\text{FONDUE}_{IB}$, we simply replace the difference between traces by the difference between IDEs as shown in Algorithms 5 and 6.

---

**Algorithm 5** $\text{FONDUE}_{IDE}$

1: **procedure** $\text{FONDUE}_{IDE}(t, IDE_{data}, e)$
2:    $l \leftarrow 0$
3:    $u \leftarrow \infty$
4:    $n \leftarrow IDE_{data}$
5:    $m \leftarrow \{\}$
6:    **while** $n \neq l$ **do**
7:       $IDE_z, IDE_\mu \leftarrow \text{GET-MEM}(m, n, e)$
8:       **if** $(IDE_z - IDE_\mu) \leqslant t$ **then**
9:          $l \leftarrow n$
10:         $n \leftarrow \min(n \times 2, u)$
11:       **else**
12:         $u \leftarrow n$
13:         $n \leftarrow \text{floor}\left(\frac{l+u}{2}\right)$
14:       **end if**
15:    **end while**
16:    **return** $n$
17: **end procedure**

**Algorithm 6** $\text{GET-MEM}_{IDE}$

1: **procedure** $\text{GET-MEM}(m, n, e)$
2:    **if** $m[n] = \emptyset$ **then**
3:       $vae \leftarrow \text{TRAIN-VAE}(dim = n, n\_epochs = e)$
4:       $IDE_z, IDE_\mu \leftarrow IDEs(vae)$
5:       $m[n] \leftarrow IDE_z, IDE_\mu$
6:    **end if**
7:    **return** $m[n]$
8: **end procedure**

---

Table 9: Number of latent variables $n$ obtained with FONDUE and FONDUE$_{IDE}$. The results are averaged over 10 seeds, and computation times are reported for NVIDIA A100 GPUs. The computation time is given for one run of the algorithm over the minimum number of epochs needed to obtain a stable score.

|  | Dataset | $n$ (avg $\pm$ SD) | Time/run | Models trained | Epochs/training |
|---|---|---|---|---|---|
| FONDUE | Symsol | $19.1 \pm 0.7$ | 6 min | 8 | 1 |
| FONDUE | dSprites | $10.9 \pm 0.7$ | 42 min | 4 | 2 |
| FONDUE | Celeba | $32.6 \pm 0.7$ | 17 min | 6 | 2 |
| FONDUE$_{IDE}$ | Symsol | $12.6 \pm 0.5$ | 10 min | 6 | 2 |
| FONDUE$_{IDE}$ | dSprites | $10.1 \pm 0.7$ | 42 min | 4 | 2 |
| FONDUE$_{IDE}$ | Celeba | $34.7 \pm 1.0$ | 32 min | 6 | 4 |

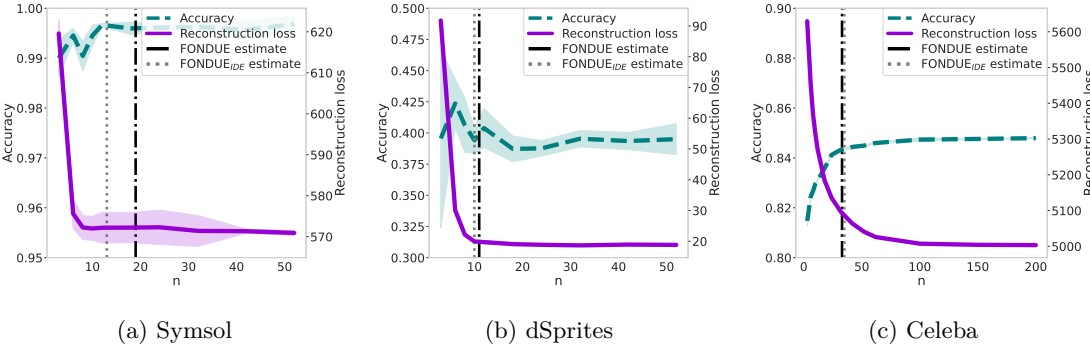

(a) Symsol  (b) dSprites  (c) Celeba

Figure 13: Reconstruction loss and accuracy obtained for generation and downstream tasks of **VAEs** for Symsol, dSprites, and Celeba with an increasing number of latent variables. The plain and dashed vertical lines indicate the number of dimensions found by FONDUE and FONDUE$_{IDE}$.

**How to select a suitable value of $t$?** As we do not have a theoretical relationship between IDE and the polarised regime, it is more complicated to provide a principled way to select the threshold of FONDUE$_{IDE}$. While $t$ was set to a fixed value of 1 as in Section 4, one could wonder if this would be a good fit for their particular use case. By looking at Figures 10 and 11, one can see that the difference between the IDEs of the mean and sampled representations is generally close to the number of additional mixed and passive variables. Thus, $t$ represents this number of "extra variables" (mixed and passive) that we want to allow the model to use, indicating that the threshold obtained for FONDUE is readily applicable to FONDUE$_{IDE}$.

**Obtaining stable estimates** As in Section 5, to ensure stable estimates, we computed FONDUE multiple times, gradually increasing the number of epochs $e$ until the predicted $p$ stopped changing. As reported in Table 9, the results were generally stable after two epochs, except for Celeba which needed four.

**Analysing the results of FONDUE** As shown in Table 9, the execution time of FONDUE$_{IDE}$ for finding the number of dimensions for one dataset is much shorter than for fully training one model (approximately 2h using the same GPUs) but generally longer than the original FONDUE algorithm. Moreover, one can see in Figure 13 that the number of latent dimensions predicted by FONDUE and FONDUE$_{IDE}$ are very close for dSprites and Celeba. FONDUE$_{IDE}$ also performs better on Symsol with a number of dimensions closer to what would be chosen with the Elbow method. We hypothesise that FONDUE$_{IDE}$ may cope better with more noisy setups where multiple runs of a VAE reach very different reconstruction loss. It could thus be interesting to investigate ID estimation through the lens of the polarised regime to provide a more robust alternative to FONDUE with theoretical guarantees.

# G   Additional results

This section provides additional observations of the IDEs of VAEs which are complementary to Appendix F but not necessary for understanding FONDUE$_{IDE}$.

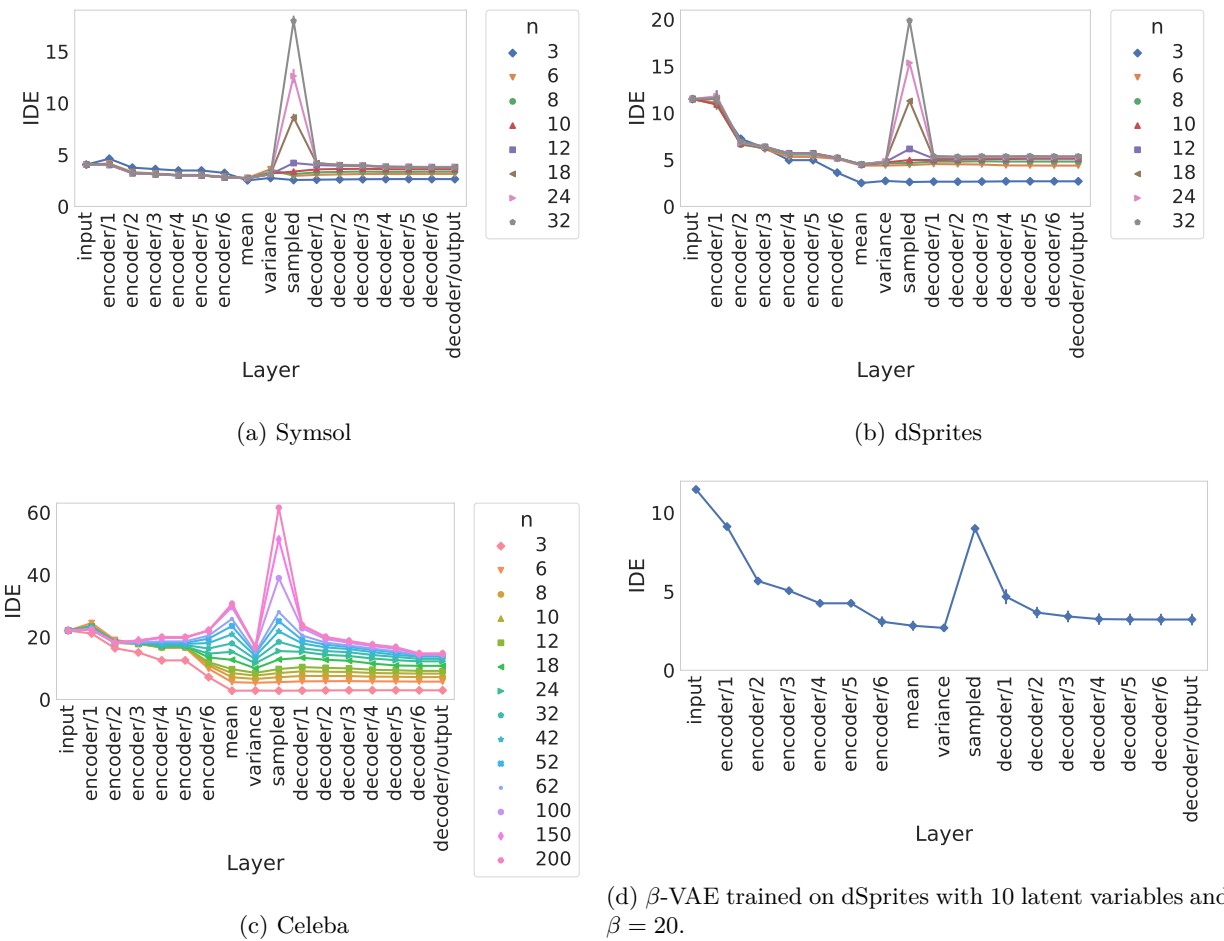

Figure 14: IDEs of VAEs trained with an increasing number of latent dimensions $n$. (a), (b), and (c) show the results on Symsol, dSprites, and Celeba, respectively. (d) shows the results of $\beta$-VAEs trained on dSprites with 10 latent variables and $\beta = 20$ to cause posterior collapse.

**What happens in the case of posterior collapse?**   By using a $\beta$-VAE with very large $\beta$ (e.g., $\beta = 20$), one can induce posterior collapse, where a majority of the latent variables become passive and prevent the decoder from accessing sufficient information about the input to provide a good reconstruction. This phenomenon is illustrated in Figure 14d, where the IDEs of the encoder representations are similar to what one would obtain for a well performing model in the first 5 layers, indicating that these early layers of the encoder still encode some useful information about the data. The IDEs then drop in the last three layers of the encoder, indicating that most variables are passive, and only a very small amount of information is retained. The IDE of the sampled representation (see *sampled* in  Figure 14d) is then artificially inflated by the passive variables and becomes very close to the number of dimensions $n$. From this, the decoder is unable to learn much and has thus a low IDE, close to the IDE of the mean representation (see the points on the RHS of Figure 14d).

**The IDEs of the encoder representations decrease, but the IDEs of the decoder representations stay constant**   We can see in Figure 14 that the IDE of the representations learned by the encoder

decreases until we reach the mean and variance layers, which is consistent with the observations reported for classification (Ansuini et al., 2019). Interestingly, for dSprites and Symsol, when the number of latent variables is at least equal to the IDE of the data, the IDE of the mean and variance representations is very close to the true data ID. After a local increase of the IDE in the sampled representations, the IDE of the decoder representations stays close to the IDE of the mean representations and does not change much between layers.

## H    Impact of the initialisation

In Section 5.2, we used the IDE of the dataset as the initial number of dimensions. To show the impact of this choice on FONDUE, we run the algorithm with 10 randomly selected initial numbers of dimensions chosen between 1 and 200 on 10 different seeds. We can see in Tables 10, 11 and 12 that the initialisation does not change the number of dimensions predicted with IDE initialisation in Table 1, but the execution time can be longer when the initial number of dimensions is further away from the predicted number of dimensions. Despite this, all these results remain below the average training time needed for one model on the same GPU (around 2 hours). To conclude, while using the data IDE as the initial number of dimensions does not change the results of FONDUE, it allows the algorithm to start closer to the predicted number of dimensions and thus can shorten its running time.

Table 10: Number of latent variables $n$ obtained with FONDUE with random initial numbers of dimensions $n_{\text{init}}$ on Symsol. The results are averaged over 10 seeds, and computation times are reported for NVIDIA A100 GPUs. The computation time is given for one run of the algorithm over the minimum number of epochs needed to obtain a stable score. The last line is the average over all initial numbers of dimensions.

| $n$ (avg $\pm$ SD) | Time/run | Models trained | Epochs/training | $n_{\text{init}}$ |
|---|---|---|---|---|
| $19.3 \pm 0.5$ | 6 min | 7 | 1 | 5 |
| $18.2 \pm 0.4$ | 4 min | 4 | 1 | 19 |
| $18.7 \pm 0.8$ | 4 min | 5 | 1 | 22 |
| $19.0 \pm 0.8$ | 5 min | 6 | 1 | 39 |
| $18.9 \pm 0.9$ | 7 min | 8 | 1 | 74 |
| $19.0 \pm 0.8$ | 8 min | 9 | 1 | 94 |
| $18.7 \pm 0.5$ | 8 min | 9 | 1 | 116 |
| $19.3 \pm 0.7$ | 7 min | 8 | 1 | 145 |
| $19.1 \pm 0.6$ | 7 min | 8 | 1 | 180 |
| $18.8 \pm 0.9$ | 8 min | 9 | 1 | 182 |
| $18.9 \pm 0.7$ | 6 min | 8 | 1 | |

## I    DIP-VAE II

In DIP-VAE II (Kumar et al., 2018), the approach is slightly different from the $\beta$-VAE objective presented in Equation 2. The authors argued that to prevent blurry output, only the distance between the estimated latent factors and the prior should be penalised, and they proposed a new objective to this end:

$$\mathbb{E}_{p(\mathbf{x})}\left[\mathbb{E}_{q_\phi(\mathbf{z}|\mathbf{x})}\left[\log p_{\boldsymbol{\theta}}\big(\mathbf{x}|\mathbf{z}\big)\right] - D_{\text{KL}}\big(q_\phi\big(\mathbf{z}|\mathbf{x}\big) \parallel p_{\boldsymbol{\theta}}(\mathbf{z})\big)\right] - \gamma D_{\text{KL}}\big(q_\phi(\mathbf{z}) \parallel p_{\boldsymbol{\theta}}(\mathbf{z})\big). \tag{35}$$

Here $D_{\text{KL}}\big(q_\phi(\mathbf{z}) \parallel p_{\boldsymbol{\theta}}(\mathbf{z})\big)$ is measured by matching the moments of the learned distribution $q_\phi(\mathbf{z})$ and its prior $p_{\boldsymbol{\theta}}(\mathbf{z})$. The second moment of the learned distribution is given by:

$$Cov_{q_\phi(\mathbf{z})}[\mathbf{z}] = Cov_{p(\mathbf{x})}[\boldsymbol{\mu}] + \mathbb{E}_{p(\mathbf{x})}[\boldsymbol{\sigma}]. \tag{36}$$

Two divergences are then defined. The first, DIP-VAE I, penalises only the first term of Equation 36:

$$\lambda D_{\text{KL}}\big(q_\phi(\mathbf{z}) \parallel p_{\boldsymbol{\theta}}(\mathbf{z})\big) = \lambda_{od} \sum_{i \neq j} \big(Cov_{p(\mathbf{x})}[\boldsymbol{\mu}]\big)_{ij}^2 + \lambda_d \sum_i \big(Cov_{p(\mathbf{x})}[\boldsymbol{\mu}]_{ii} - 1\big)^2,$$

Table 11: Number of latent variables $n$ obtained with FONDUE with random initial numbers of dimensions $n_{\text{init}}$ on Celeba. The results are averaged over 10 seeds, and computation times are reported for NVIDIA A100 GPUs. The computation time is given for one run of the algorithm over the minimum number of epochs needed to obtain a stable score. The last line is the average over all initial numbers of dimensions.

| $n$ (avg $\pm$ SD) | Time/run | Models trained | Epochs/training | $n_{\text{init}}$ |
|---|---|---|---|---|
| $32.4 \pm 1.9$ | 23 min | 8 | 2 | 5 |
| $32.3 \pm 0.8$ | 20 min | 7 | 2 | 19 |
| $32.4 \pm 1.2$ | 17 min | 6 | 2 | 22 |
| $31.6 \pm 1.8$ | 23 min | 8 | 2 | 39 |
| $31.7 \pm 0.8$ | 23 min | 8 | 2 | 74 |
| $31.9 \pm 1.1$ | 26 min | 9 | 2 | 94 |
| $32.2 \pm 0.4$ | 23 min | 8 | 2 | 116 |
| $32.1 \pm 1.1$ | 23 min | 8 | 2 | 145 |
| $32.2 \pm 1.3$ | 26 min | 9 | 2 | 180 |
| $31.9 \pm 1.3$ | 28 min | 10 | 2 | 182 |
| $32.1 \pm 1.2$ | 28 min | 8 | 2 | |

Table 12: Number of latent variables $n$ obtained with FONDUE with random initial numbers of dimensions $n_{\text{init}}$ on dSprites. The results are averaged over 10 seeds, and computation times are reported for NVIDIA A100 GPUs. The computation time is given for one run of the algorithm over the minimum number of epochs needed to obtain a stable score. The last line is the average over all initial numbers of dimensions.

| $n$ (avg $\pm$ SD) | Time/run | Models trained | Epochs/training | $n_{\text{init}}$ |
|---|---|---|---|---|
| $10.6 \pm 1.1$ | 63 min | 6 | 2 | 5 |
| $10.6 \pm 0.8$ | 63 min | 6 | 2 | 19 |
| $10.7 \pm 0.5$ | 53 min | 5 | 2 | 22 |
| $11.2 \pm 0.6$ | 74 min | 7 | 2 | 39 |
| $11.2 \pm 0.8$ | 74 min | 7 | 2 | 74 |
| $11.1 \pm 1.0$ | 84 min | 8 | 2 | 94 |
| $10.5 \pm 1.1$ | 84 min | 8 | 2 | 116 |
| $11.0 \pm 0.7$ | 84 min | 8 | 2 | 145 |
| $10.9 \pm 0.6$ | 84 min | 8 | 2 | 180 |
| $11.1 \pm 0.9$ | 84 min | 8 | 2 | 182 |
| $10.9 \pm 0.8$ | 75 min | 7 | 2 | |

where $\lambda_{od}$ and $\lambda_d$ are the off-diagonal and diagonal regularisation terms, respectively. The second, DIP-VAE II, penalises both terms of Equation 36:

$$\lambda D_{\mathrm{KL}}\big(q_\phi(\mathbf{z}) \parallel p_\theta(\mathbf{z})\big) = \lambda_{od} \sum_{i \neq j} \big(Cov_{q_\phi(\mathbf{z})}[\mathbf{z}]\big)_{ij}^2 + \lambda_d \sum_i \big(Cov_{q_\phi(\mathbf{z})}[\mathbf{z}]_{ii} - 1\big)^2.$$

In practical terms, it means that DIP-VAE I enforces the covariance matrix of the mean representation to be diagonal, while DIP-VAE II explicitly regularises the covariance matrix of the sampled representation. While the difference may seems minor, it was shown that only enforcing diagonal covariance of the sampled representation can result in discrepancies between the mean and sampled representations which are not observed in DIP-VAE I (Locatello et al., 2019b).

## J Additional related work

**Elbow method based on the Structure Preservation Index** In the context of deterministic AEs trained on textual data, Gupta et al. (2016) proposed to apply the Elbow method to the Structure Preservation Index (SPI) instead of the reconstruction loss. The idea of SPI is to capture structural distortions between the input documents and their reconstruction. It is defined as follows:

$$SPI = \frac{1}{h} \sum_{i,j} \|\boldsymbol{D}_{ij} - \hat{\boldsymbol{D}}_{ij}\|, \tag{37}$$

where $\boldsymbol{D}_{ij}$ is the cosine similarity between the documents $\mathbf{x}^{(i)}$ and $\mathbf{x}^{(j)}$, and $\hat{\boldsymbol{D}}_{ij}$ is calculated the same way with the reconstructed documents $\hat{\mathbf{x}}^{(i)}$ and $\hat{\mathbf{x}}^{(j)}$.

**Human supervision based on the information plane** Yu & Príncipe (2019) studied the information of the input preserved by the bottleneck layer $\mathbf{z}$ of stacked AEs (Vincent et al., 2010) using the information plane spanned by $\mathrm{I}(\mathbf{x}, \mathbf{z})$ and $\mathrm{I}(\hat{\mathbf{x}}, \mathbf{z})$, where $\mathrm{I}(\cdot, \cdot)$ denotes mutual information. In order to approximate the true entropy, they used a kernel estimator of the Rényi's $\alpha$-order entropy. Specifically, given the samples $\boldsymbol{X} = \big\{\mathbf{x}^{(i)}\big\}_{i=1}^h$ of a random variable $\mathbf{x}$, a positive definite kernel $\kappa$, the resulting Gram matrix $\boldsymbol{K}$ where $\boldsymbol{K}_{i,j} \triangleq \kappa(\mathbf{x}^{(i)}, \mathbf{x}^{(j)})$, and its normalised version $\boldsymbol{A}_{i,j} \triangleq \frac{\boldsymbol{K}_{i,j}}{h\sqrt{\boldsymbol{K}_{i,i}\boldsymbol{K}_{j,j}}}$, the entropy estimator is

$$S_\alpha(\boldsymbol{A}) \triangleq \frac{1}{1-\alpha} \log_2 \left( \sum_{i=1}^h \lambda_i(\boldsymbol{A})^\alpha \right), \tag{38}$$

where $\lambda_i(A)$ denotes the i$^{\mathrm{th}}$ eigenvalue of $\boldsymbol{A}$.

Similarly, the joint entropy is estimated by

$$S_\alpha(\boldsymbol{A}, \boldsymbol{B}) \triangleq S_\alpha \left( \frac{\boldsymbol{A} \odot \boldsymbol{B}}{\mathrm{Tr}(\boldsymbol{A} \odot \boldsymbol{B})} \right), \tag{39}$$

where $\odot$ denotes the Hadamard product,' and $A$ and $B$ are normalised Gram matrices as before. From Equations 38 and 39, one can thus obtain the mutual information,

$$\mathrm{I}_\alpha(\boldsymbol{A}, \boldsymbol{B}) \triangleq S_\alpha(\boldsymbol{A}) + S_\alpha(\boldsymbol{B}) - S_\alpha(\boldsymbol{A}, \boldsymbol{B}). \tag{40}$$

In their experiment, Yu & Príncipe (2019) set $\alpha = 1.01$ and chose a Radial Basis Function (RBF) kernel, such that, given $\boldsymbol{X} \in \mathbb{R}^{h \times m}$

$$\kappa(\mathbf{x}^{(i)}, \mathbf{x}^{(j)}) = \exp \left( -\frac{\|\mathbf{x}^{(i)} - \mathbf{x}^{(j)}\|^2}{2s^2} \right), \tag{41}$$

where $\|\cdot\|^2$ is the squared Euclidean distance. $s$ is estimated based on Silverman's rule of thumb for Gaussian density estimation (Silverman, 1998), such that $s \triangleq \sigma(\boldsymbol{X})h^{-1/(4+m)}$ where $\sigma$ denotes the standard deviation. The authors observed that for a large enough $n$, the information plane started to display curved patterns and concluded that a good number of latent dimensions $n$ corresponded to the information plane just before the appearance of this change of pattern, which can be seen as an application of the Elbow method. The proposed technique requires to fully train multiple models and visually inspect the information planes obtained for different $n$.

# K   Comparison in the supervised setting

In this section, we compare the results obtained by the IB algorithm with the results obtained with $\mathrm{Tr}(\mathrm{Var}[\boldsymbol{Z}]) - \mathrm{Tr}(\mathrm{Var}[\boldsymbol{M}])$ in the supervised setting. To avoid unfair comparison due to IB fully training one or more models depending on the chosen range, we restrict both implementations to the number of epochs after which they provide stable estimate for each dataset, as per Table 1. The implementations of the IB algorithm and the supervised version of $\mathrm{Tr}(\mathrm{Var}[\boldsymbol{Z}]) - \mathrm{Tr}(\mathrm{Var}[\boldsymbol{M}])$, $\mathrm{BS}_{FONDUE}$, are shown in Algorithms 7 and 8. We further define $l = 1$ and $u = 200$ for all the runs. Note that the GET-MEM functions are the same as Algorithms 2 and 4 and the thresholds $t$ remain unchanged. For IB, Algorithm 7 is thus equivalent to the original implementation of (Boquet et al., 2021) except that any model training is fixed to a given number of epochs. We can see in Table 13 that for the chosen range, IB and $\mathrm{BS}_{FONDUE}$ take longer to compute than FONDUE and $\mathrm{FONDUE}_{IB}$ for all datasets except Symsol where the execution time is similar. Indeed, the number of models to partially train is higher with binary search than in the unsupervised setting for dSprites and Celeba but similar for Symsol. The predicted number of dimensions are consistent with Table 1, $\mathrm{BS}_{FONDUE}$ being closer to the results obtained with Elbow methods than IB, as before.

---

**Algorithm 7** IB

1: **procedure** $\mathrm{IB}(t, l, u, e)$
2:     $m \leftarrow \{\}$
3:     **while** $l < u$ **do**
4:         $n \leftarrow \mathrm{floor}\left(\frac{l+u}{2}\right)$
5:         $\mathrm{H}(\mathbf{z}),\ \mathrm{I}(\mathbf{x}, \hat{\mathbf{x}}) \leftarrow \mathrm{GET\text{-}MEM}_{\mathrm{IB}}(m, n, e)$
6:         **if** $\mathrm{H}(\mathbf{z}) - \mathrm{I}(\mathbf{x}, \hat{\mathbf{x}}) \leqslant t$ **then**
7:             $l \leftarrow n + 1$
8:         **else**
9:             $u \leftarrow n$
10:        **end if**
11:     **end while**
12:     **return** $l - 1$
13: **end procedure**

---

**Algorithm 8** $\mathrm{BS}_{FONDUE}$

1: **procedure** $\mathrm{BS}_{FONDUE}(t, l, u, e)$
2:     $m \leftarrow \{\}$
3:     **while** $l < u$ **do**
4:         $n \leftarrow \mathrm{floor}\left(\frac{l+u}{2}\right)$
5:         $\mathrm{Tr}(\mathrm{Var}[\boldsymbol{Z}]),\ \mathrm{Tr}(\mathrm{Var}[\boldsymbol{M}]) \leftarrow \mathrm{GET\text{-}MEM}(m, n, e)$
6:         **if** $\mathrm{Tr}(\mathrm{Var}[\boldsymbol{Z}]) - \mathrm{Tr}(\mathrm{Var}[\boldsymbol{M}]) \leqslant t$ **then**
7:             $l \leftarrow n + 1$
8:         **else**
9:             $u \leftarrow n$
10:        **end if**
11:     **end while**
12:     **return** $l - 1$
13: **end procedure**

---

Table 13: Number of latent variables $n$ obtained with $\text{BS}_{FONDUE}$ and IB. The results are averaged over 10 seeds, and computation times are reported for NVIDIA A100 GPUs. The computation time is given for one run of the algorithm over the minimum number of epochs needed to obtain a stable score.

|  | Dataset | $n$ (avg $\pm$ SD) | Time/run | Models trained | Epochs/training |
|---|---|---|---|---|---|
| $\text{BS}_{FONDUE}$ | Symsol | $19.2 \pm 0.6$ | 6 min | 8 | 1 |
| $\text{BS}_{FONDUE}$ | dSprites | $11.3 \pm 0.8$ | 75 min | 8 | 2 |
| $\text{BS}_{FONDUE}$ | Celeba | $32.2 \pm 1.0$ | 21 min | 8 | 2 |
| IB | Symsol | $23.6 \pm 0.8$ | 40 min | 8 | 6 |
| IB | dSprites | $15.6 \pm 0.5$ | 74 min | 8 | 2 |
| IB | Celeba | $18.5 \pm 0.5$ | 31 min | 8 | 3 |

# L    Generalisation to different learning objectives and hyperparameter values

This section provides additional figures comparing $\beta$-VAEs with different $\beta$ values and DIP-VAEs II with different $\lambda_{od}$ values with Elbow methods. Overall, we can see in Figures 15 to 18 that FONDUE provides consistent results across hyperparameter values and learning objectives, with some overestimation for Symsol as observed in Section 5.2.

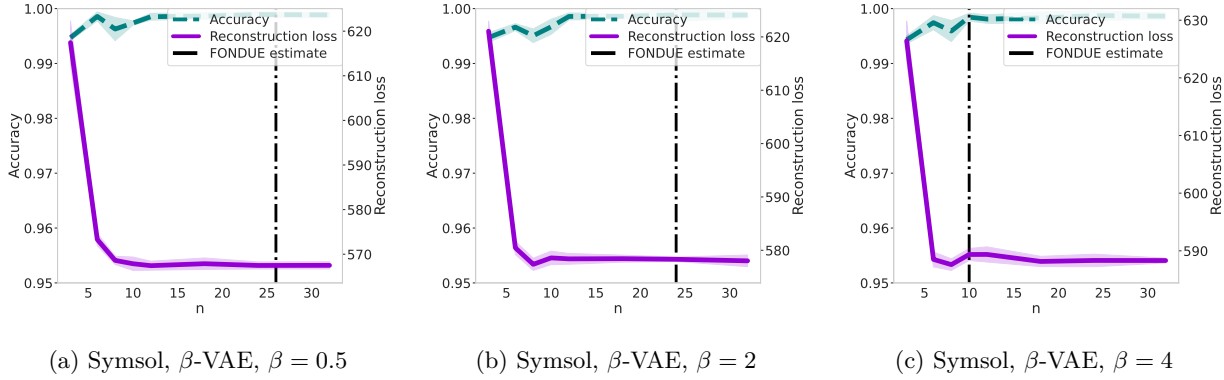

(a) Symsol, $\beta$-VAE, $\beta = 0.5$       (b) Symsol, $\beta$-VAE, $\beta = 2$       (c) Symsol, $\beta$-VAE, $\beta = 4$

Figure 15: Reconstruction loss and accuracy obtained for generation and downstream tasks with an increasing number of latent variables on Symsol with $\beta$-VAE. (a) shows the results for $\beta = 0.5$, (b) for $\beta = 2$, and (c) for $\beta = 4$

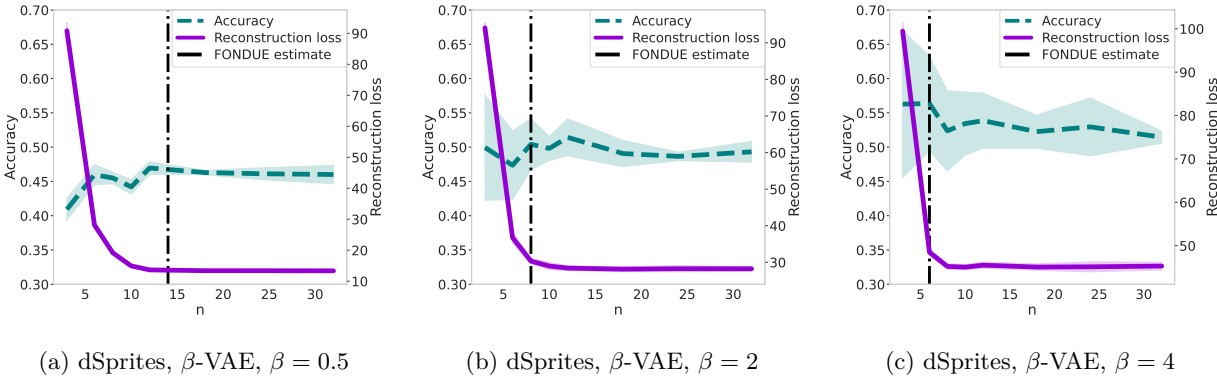

(a) dSprites, $\beta$-VAE, $\beta = 0.5$       (b) dSprites, $\beta$-VAE, $\beta = 2$       (c) dSprites, $\beta$-VAE, $\beta = 4$

Figure 16: Reconstruction loss and accuracy obtained for generation and downstream tasks with an increasing number of latent variables on dSprites with $\beta$-VAE. (a) shows the results for $\beta = 0.5$, (b) for $\beta = 2$, and (c) for $\beta = 4$

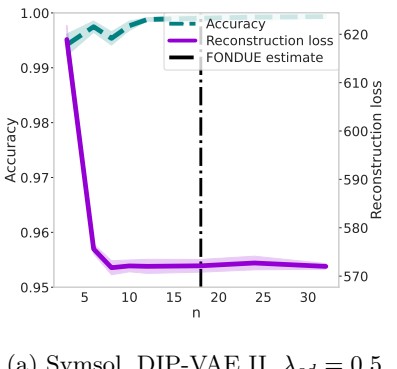 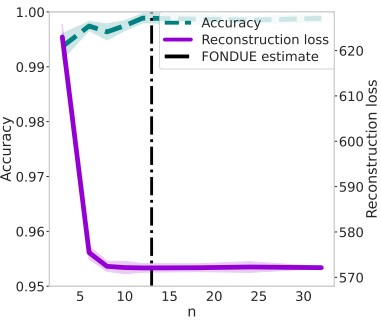

(a) Symsol, DIP-VAE II, $\lambda_{od} = 0.5$        (b) Symsol, DIP-VAE II, $\lambda_{od} = 2$

Figure 17: Reconstruction loss and accuracy obtained for generation and downstream tasks with an increasing number of latent variables on Symsol with DIP-VAE II. (a) shows the results for $\lambda_{od} = 0.5$ and (b) for $\lambda_{od} = 2$.

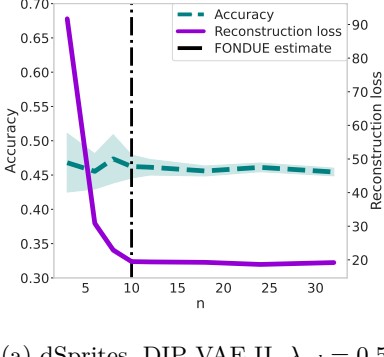 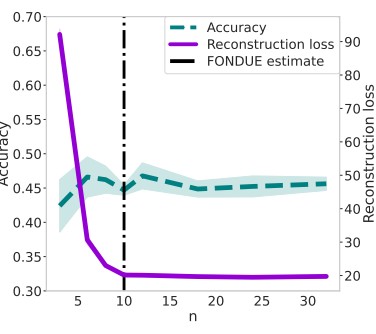

(a) dSprites, DIP-VAE II, $\lambda_{od} = 0.5$        (b) dSprites, DIP-VAE II, $\lambda_{od} = 2$

Figure 18: Reconstruction loss and accuracy obtained for generation and downstream tasks with an increasing number of latent variables on dSprites with DIP-VAE II. (a) shows the results for $\lambda_{od} = 0.5$ and (b) for $\lambda_{od} = 2$.

