# OpenReview forum: "FONDUE: an algorithm to automatically find the dimensionality of the latent representations of variational autoencoders"
_TMLR — Rejected by TMLR_

### Review · Reviewer_6PWv · 2023-07-21

**Summary Of Contributions:**

The paper introduces a method to estimate the dimensionality of the latent space in variational autoencoders, based on detecting the presence of passive or mixed latent variables.

**Audience:**

Yes

**Broader Impact Concerns:**

No concerns.

**Claims And Evidence:**

No

**Requested Changes:**

**Critical issues:**

The following aspects are not explained clearly in the paper:
* I think it is important to explain clearly what _"mean and sampled representations"_ are. This is assumed to be self-evident throughout the paper. Including a simple explanation in the Introduction, and a technical one later, would improve the readability.
* On page two, several symbols are introduced without being previously defined, including $\boldsymbol{\epsilon}$. It would be better if it were specified what this symbol means before introducing it.
* Same for $p(\boldsymbol{\mu})$, $p(\boldsymbol{\epsilon})$ etc. on page 3
* In section 2.4, the authors write: _", we assume that the considered models are learning under the polarised regime [...] this requirement can be made without loss of generality as it has been shown that the polarised regime was necessary for VAEs to learn properly"_. I find this statement unclear or misleading: "without loss of generality" with respect to what? The polarised regime only occurs for certain hyperparameter settings of VAEs, not all, which seems to imply that there _is_ loss of generality in assuming it throughout. Rather, I would rephrase as something like: _"our work is based on assumptions which hold for many realistic hyperparameter settings in VAEs"_

**Minor issues:**
* The long and detailed section 2.5 on Related work does not appear to be crucial to the rest of the paper (possibly apart from the IB algorithm), and impairs readability. My suggestion would be to shorten it significantly, or to move to the end of the paper or in the Appendix.
* In practice, the method requires training VAEs sequentially, for few epochs, and updating $n$. Is it clear that it works better and faster than methods where ensembles of VAEs can be trained in parallel (possibly also for few epochs each)?
* I am confused by the paragraph "Can FONDUE be applied to other architectures and learning objectives?". It is unclear to me what mean and sampled representations are for (deterministic) autoencoders, and I did not find that this was explained clearly.

**Strengths And Weaknesses:**

**Strenghts:**

* The idea underlying the proposed method is simple and intuitively appealing.

**Weaknesses:**

* Many results lack precise formalization. Notably, the $\approx$ sign is used throughout to represent an approximate equality. This notion of approximate equality seems imprecise: as a result, most of the theoretical results lean towards qualitative rather than quantitative statements. One possibility to address this could be to adjust (reduce) some of the claims. I try to provide some suggestions in this direction below, in the "Requested Changes".
* The method appears to rely on a number of assumptions (e.g., that active variables already appear _"after a few epochs"_) and on a particular choice of hyperparameters (e.g., the $\beta$ hyperparameter), which limits its generality. Moreover, experimental verification of the method's robustness to different hyperparameter settings is lacking in the main paper.
* From the Background: _"[...] we will use Intrinsic Dimension Estimates (IDEs) as the initial number of dimensions n for FONDUE"_. Based on this sentence, it is unclear whether some of the statements following in the paper are fully justified. For example, at the end of section 2: _"our approach is **fully unsupervised**, while the IB algorithm requires human supervision to select a range of likely latent dimensions"_ (bold mine). On the other hand, if FONDUE relies on a good initialization of the number of dimensions _n_, this also constitutes, in some sense, a form of supervision, and it should be acknowledged in the paper.

---

> ### Author Response · Authors · 2023-09-06
> **Answer to reviewer 6PWv (1/3)**
>
> Thank you for your interest in our work and for your detailed comments.
>
> Answers to Reviewer's Questions:
> --------------------------------
>
> > Many results lack precise formalization [...] suggestions in this direction below, in the "Requested Changes".
>
> The $\approx$ sign used in the original definition of the polarised regime [1] comes from the fact that an equality
> would only be reached for a decoder with a variance of 0 [2]. We have implemented the suggested changes
> (see below) to address this issue. Furthermore, as suggested by reviewer 1dXF, we proposed an alternative reformulation
> of our definitions and propositions, expressing them in terms of limits and removing any approximations.
>
> > The method appears to rely on a number of assumptions [...] which limits its generality.
>
> We have now added mode details in the paragraph *Assumptions* to clarify what the assumptions are and what they entail in terms of generalisation.
> In the paragraph *Can FONDUE be applied to other architectures and learning objectives?* we now summarise how well FONDUE
> generalises on multiple models and different values of $\beta$.
>
> *About the quick convergence assumption*:
> Indeed, to give meaningful results, Theorems 1 and 2 need to have mean and sampled representations which reflect
> accurately the number of active and passive variables present in the final model. [1,2,3] observed that variables of the mean and sampled representations
> are characteristic of the polarised regime after a few epochs. Thus, using the observations of [1,2,3] we can say that we will have access to mean and sampled
> representations which are sufficiently stable for Theorems 1 and 2 to provide consistent results very early in the training.
>
> To clarify this point, we have reformulated
>
> ``A useful property of Theorems 1 and 2 is that they hold very early in training (Bonheme & Grzes, 2022).``
>
> as
>
> ``A useful property of VAEs is that their mean and sampled representations converge very early in the training.
> Indeed, the variance of the decoder very quickly approaches $0$ [2,3] leading to observations
> of the polarised regime after a few epochs [1,4].
> This allows Theorem 1 and 2 to provide stable estimates using mean and sampled representations which reflect accurately
> the number of active and passive variables present in the final model.``
>
> Moreover, VAEs whose latent space does not converge quickly will suffer from suboptimal performance due to lagging inference [5].
> Thus, the assumption that active variables appear ``after a few epochs'' is only restricting the algorithm to well
> performing VAEs (i.e., without lagging inference), which means that this is not a very limiting assumption.
> This is also observed empirically in Tables 1 and 6 whose results are consistently stable after less than 5 epochs.
> We agree however that if a model required a large number of epochs for active and passive variables to appear, this
> would make FONDUE much slower and we emphasised this fact in the *Assumption* paragraph.
>
> *Hyperparameter choice*:
> We have now added experiments with another VAE model: DIP-VAE II [6].
> For $\beta$-VAE and DIP-VAE II, we have also added results obtained on a range of hyperparameter values.
> $\beta = [0.5, 2, 4]$ for beta VAE and $\lambda_{od} = [0.5, 1, 2, 4]$ for DIP VAE II.
> These results are summarised in the *Can FONDUE be applied to other architectures and learning objectives?* paragraph
> and in Appendix L. Note that the current version does not contain the graphs for Celeba and some values of
> $\lambda_{od}$ for which models still being trained. We will add them to Appendix L as soon as they are available.
>
>
> > Moreover, experimental verification of the method's robustness [...] in the main paper.
>
> This is now addressed by the additional experiments and the *Can FONDUE be applied to other architectures and learning objectives?*
> paragraph described above as well as in Appendix L.
>
>
> > From the Background [...] it should be acknowledged in the paper.
>
> We have clarified this by replacing "our approach is fully unsupervised, while the IB algorithm requires human supervision to select a range of likely latent dimensions" by "the IB algorithm requires human supervision to select a range of likely latent dimensions which is not needed by FONDUE". Because FONDUE provably converges, we could choose any integer as an initial value.
> We have added an experiment using 10 random initial numbers of dimensions chosen between 1 and 200 to show this phenomenon in Appendix H and included a summary of these results in the paragraph *How does FONDUE work?*. Our experimental results confirm that the initialisation value does not impact the number of dimensions predicted by FONDUE. When the initial number of dimensions is further away from the predicted value, the algorithm may take longer to converge as it may need more iterations (this is the case for Celeba and dSprites for example) but we still are below the time required to fully train one model.

---

> > ### Author Response · Authors · 2023-09-06
> > **Answer to reviewer 6PWv (2/3)**
> >
> > > I think it is important to explain clearly [...] would improve the readability.
> >
> > We have added a schema explaining what the mean and sampled representations are as well as an explanation in the introduction and background as suggested.
> >
> > > On page two, several symbols are introduced [...] means before introducing it. Same for [...] page 3
> >
> > This is now fixed.
> >
> > > In section 2.4, the authors write [...] realistic hyperparameter settings in VAEs"
> >
> > Thank you for the suggestion, we have rephrased this sentence accordingly.
> > We have also added some explanation about which VAEs learn in a polarised regime in the *Assumptions* paragraph described above.
> >
> > > The long and detailed section 2.5 on Related work [...] the Appendix.
> >
> > As suggested, we have moved any non-essential related work to Appendix J and kept only the content relevant to the understanding of the elbow methods and IB algorithm.
> >
> > > In practice, the method requires training VAEs sequentially [...] (possibly also for few epochs each)?
> >
> > We have seen in Figures 5-6 that FONDUE results match the Elbow, which is the best trade-off we can get between reconstruction
> > quality and downstream task accuracy. So in that sense, FONDUE does not provide a ''better'' estimate than the (gold standard) Elbow method. However, it does not require extensive full training of a range of VAEs with different numbers of dimensions nor to specify manually (and possibly incorrectly) a range of dimensionalities to inspect. So, in that sense, FONDUE is ''better'' because one can obtain results close to the gold standard in a few minutes in an unsupervised way (with the only inductive bias of the initialisation using IDE as mentioned before).
> >
> > As discussed above, FONDUE also has the additional benefit of not needing any direct human supervision (the only inductive bias being the IDE initialisation, which may, in some cases like Symsol, be far from the target value as the IDE of 4 is far away from the 19 dimensions predicted, but FONDUE is guaranteed to converge to the correct value).
> >
> > Regarding whether it is faster than other algorithms, it is definitely faster than Elbow methods (even run in parallel) as they require full training of the VAE models.
> > The IB algorithm could be faster using parallel binary search on a carefully selected range of values (see the discussion with reviewer 1dXF and zDJy about this), but we have seen that the IB algorithm provides results that are further away from the gold standard (Elbow methods).
> > For completeness, we now compare the results obtained with a supervised version with a fixed number of epochs (to avoid biasing the results by fully training some models).
> > We have added a comparison between the IB algorithm (trained for a fixed number of epochs sufficient to reach stable results) and the proposed difference between traces combined with a binary search in Appendix K.
> >
> > > I am confused by the paragraph "Can FONDUE be applied to other architectures and learning objectives?" [...] was explained clearly.
> >
> > As you rightly point out, one cannot directly run FONDUE on deterministic AEs as they do not have mean and sampled representations and do not behave (as far as we know)
> > in a polarised regime. What we did for the deterministic AEs was to reuse the number of dimensions predicted by FONDUE on VAEs with the same architecture and verify whether this
> > also corresponded to the Elbow on deterministic AEs. Because the number of dimensions obtained for VAEs also matches the Elbow point of AEs, we concluded that one
> > could use the VAE results of FONDUE for deterministic AEs.
> > We realise now that the title of the paragraph can be misleading and put the results obtained for deterministic AEs
> > in a new paragraph *Are the results obtained with FONDUE on VAEs applicable to deterministic AEs?*

---

> > > ### Author Response · Authors · 2023-09-06
> > > **Answer to reviewer 6PWv (3/3)**
> > >
> > > References
> > > ---------------
> > > [1] Rolinek, M., Zietlow, D. and Martius, G. (2019). Variational Autoencoders Pursue PCA Directions (by Accident). In Proceedings of the IEEE/CVF Conference on Computer Vision and Pattern Recognition (CVPR)
> > >
> > > [2] Dai, B. and Wipf, D. (2018). Diagnosing and Enhancing VAE Models. In International Conference on Learning Representations, vol. 6.
> > >
> > > [3] Bonheme, L., & Grzes, M. (2022). How do variational autoencoders learn? insights from representational similarity. arXiv preprint arXiv:2205.08399.
> > >
> > > [4] Sungyeop Lee and Junghyo Jo. Information Flows of Diverse Autoencoders. Entropy, (7), 2021. doi: 10.3390/e23070862.
> > >
> > > [5] He, J., Spokoyny, D., Neubig, G. and Berg-Kirkpatrick, T. (2019). Lagging Inference Networks and Posterior Collapse in Variational Autoencoders. In International Conference on Learning Representations, vol. 7.
> > >
> > > [6] Kumar, A., Sattigeri, P. and Balakrishnan, A. (2018). Variational Inference of Disentangled Latent Concepts from Unlabeled Observations. In International Conference on Learning Representations, vol. 6.
> > >
> > > [7] Khemakhem, I., Kingma, D., Monti, R. and Hyvarinen, A. (2020). Variational Autoencoders and Nonlinear ICA: A Unifying Framework. In Proceedings of the Twenty Third International Conference on Artificial Intelligence and Statistics, Proceedings of Machine Learning Research, vol. 108.

---

### Review · Reviewer_1dXF · 2023-07-26

**Summary Of Contributions:**

This paper introduced an algorithm to determine the intrinsic dimensionality of variational auto-encoders. Through a binary search algorithm on some metrics on the latent representations, the proposed method can make an estimation on the intrinsic dimensionality. Experiment results showed the proposed method outperformed some existing method on real datasets.

**Audience:**

Yes

**Claims And Evidence:**

No

**Requested Changes:**

1. Remove all approximations in all definitions, propositions and theorems so that the paper is mathematically sound.

2. Be clear about the contribution of this paper with respect to existing work.

3. Relax Definition 1 to be more realistic in real datasets.

4. Make more comprehensive comparisons on the proposed method compared with baseline algorithms.

**Strengths And Weaknesses:**

Strengths:

1. The question that the authors studied is interesting.

2. The proposed method is easy to understand and easy to implement.

Weaknesses:

1. The theorems, propositions and definitions from this paper are mostly not mathematically sound. The major reason is that many assertions are made with the approximation sign $\approx$, which does not give any accurate mathematical conclusion. Take Proposition 2 as an example, the authors mention $\text{Var}[\mathbb z]-\text{Var}[\mathbb \mu]\approx s$, but we really have no information on what $\text{Var}[\mathbb z]-\text{Var}[\mathbb \mu]$ really is. To make things mathematically sound, the authors need to have some range on the approximation. Otherwise, such proposition gives us no information.

2. Definition 1 (which is a core definition in this paper) is too strong. If we have a set of mixed variables, it's not necessary that $p(\mathbb z_j)=cp(\mathbb \epsilon_j)+(1-c)p(\mathbb \mu_j)$. These two values might be close to each other, but not necessarily the same. Further more, the authors assume all data points to fall into one of the 3 cases as in this definition, which is definitely too strong.

3. The contribution of this paper is not clear. On page 6, the authors mention that Theorems 1 and 2 hold very early in training by citing paper [1]. So seems like Theorems 1 and 2 were already known in paper [1]. In this case, the major contribution of this paper will be only combining these two theorems into Theorem 3 and perform a binary search on that, which (in my opinion) is not sufficiently significant for a paper to be published at TMLR.

4. In the experiments, the authors first mention that there algorithm can not be compared directly with the IB algorithm, and provide some reasons. Some of these reasons are not so reasonable to me. For example, the authors mention that IB preforms binary search on a user-defined array and its execution time is hard to say. But I don't think a $O(\log N)$ algorithm can take a very long time as long as we can store the array in memory or disk.

5. This paper purely focuses on factorized normal latent distributions. But there are many other choices for the latent distributions for VAEs. For example, the covariance can be non-diagonal or the distribution can be non-Gaussian.

6. The paper claims that: one advantage of their algorithm is that they do not need a full training on VAEs. But the authors failed to analyze how to choose the number of epochs that are needed. Furthermore, it is interesting to see the results of existing methods if we do not do a full VAE training.

References:

[1] Lisa Bonheme and Marek Grzes. How do variational autoencoders learn? insights from representational
similarity. arXiv e-prints, 2022.

---

> ### Author Response · Authors · 2023-09-06
> **Answer to reviewer 1dXF (1/3)**
>
> Thank you for your interest in our work and for your detailed comments.
>
> Answers to Reviewer's Questions:
> --------------------------------
> > The theorems, propositions and definitions [...] which is definitely too strong.
> >
> > Remove all approximations in all definitions, propositions and theorems so that the paper is mathematically sound
> >
> > Relax Definition 1 to be more realistic in real datasets.
>
> These approximations stem from the approximations ($\approx$ and $\ll$) used in Definition 1 of the seminal work of [1].
> The other line of work exploring the polarised regime [2,3] instead assumes a decoder such that $p(\mathbf{x}|\mathbf{z})$ is the pdf of $\mathcal{N}(\bar{\mathbf{\mu}}, \lambda I)$ and
> study the phenomenon when $\lambda \to 0$. In the linear case, where the decoder distribution is $\mathcal{N}(W\mathbf{x} + \mathbf{b}, \lambda I)$, given the SVD of
> $W=\boldsymbol{U} \Lambda \boldsymbol{V}^T$ they show that the optimal variance representation is
> $$\boldsymbol{\Sigma} = P \text{diag} \left[ \frac{1}{\frac{\Lambda_{11}^2}{\lambda} + 1}, ..., \frac{1}{\frac{\Lambda_{aa}^2}{\lambda} + 1}, 1, ..., 1 \right] P^T,$$
> where $P$ is a permutation matrix.
> We can directly see that the active variables of the variance representation go to 0 when $\lambda \to 0$. The passive variables are instead equal to 1.
> A similar equality can be derived for the optimal mean representation:
> $$\mathbf{\mu} = P \text{diag}\left[\frac{\Lambda_{11}}{\Lambda_{11}^2 + \lambda}, ..., \frac{\Lambda_{aa}}{\Lambda_{aa}^2 + \lambda}, 0, ..., 0 \right] \boldsymbol{U}^T(\mathbf{x} - \mathbf{b}),$$
> where we can see that the passive variables of the mean representations tend to 0 near the optimal solution.
> They extended these results to the non-linear case using a Taylor expansion near $\mathbf{z} = \mathbf{\mu}$ and found similar results.
>
> We have now updated the definitions, propositions, theorems and related proofs following the same principle as [2,3] and re-expressed our results in terms of limits as $\lambda \to 0$.
> Moreover, [2,3] have shown that the variance of the decoder drops very early in the training, thus we can obtain a
> reasonably good approximation of these results after a few epochs.
>
> Furthermore, as requested by reviewer 6PWv, we have added additional experiments to show that FONDUE performs well
> on different models (as long as they learn in a polarised regime) and further clarify the impact of the different
> assumptions made for FONDUE.
>
>
> > The contribution of this paper is not clear [...] significant for a paper to be published at TMLR.
> >
> >Be clear about the contribution of this paper with respect to existing work.
>
> Theorems 1 and 2 were not known in [4] which discuss the representational similarity of a range of VAEs.
> We used the observations of [4] to show that Theorems 1 and 2 can be used very early in the training.
> To be more coherent with the updated theorems, propositions and definitions, we now additionally refer the reader to [2,3]
> regarding this phenomenon.
>
> As discussed above, to give meaningful results, Theorems 1 and 2 need to have mean and sampled representations which reflect
> accurately the number of active and passive variables present in the final model. What [2,3,4] tell us is that variables of the mean and sampled representations
> are characteristic of the polarised regime after a few epochs. Thus, using the observations of [2,3,4] we can say that we will have access to mean and sampled
> representations which are sufficiently stable for Theorems 1 and 2 to provide consistent results very early in the training.
>
> To clarify this point, we have reformulated
>
> ``A useful property of Theorems 1 and 2 is that they hold very early in training (Bonheme & Grzes, 2022).``
>
> as
>
> ``A useful property of VAEs is that their mean and sampled representations converge very early in the training.
> Indeed, the variance of the decoder very quickly approaches $0$ [2,3] leading to observations
> of the polarised regime after a few epochs [1,4].
> This allows Theorem 1 and 2 to provide stable estimates using mean and sampled representations which reflect accurately the number of active and passive variables present in the final model.``
>
> Moreover, we now explain the impact of the quick convergence assumption on FONDUE in the **Assumptions** paragraph mentioned above.

---

> > ### Author Response · Authors · 2023-09-06
> > **Answer to reviewer 1dXF (2/3)**
> >
> > > In the experiments, [...] in memory or disk.
> > >
> > > Make more comprehensive comparisons on the proposed method compared with baseline algorithms.
> >
> > We agree that it would be interesting to compare the results obtained with a supervised version with a fixed number of epochs (to avoid biasing the results by fully training some models).
> > We have thus added a comparison between the IB algorithm (trained for a fixed number of epochs sufficient to reach stable results) and the proposed difference between traces combined with a
> > binary search in Appendix K.
> >
> > Regarding the computational time of the IB algorithm, as it is in [5], it is hard to compare with FONDUE as:
> > - The original IB algorithm is based on a manual selection of a range of values to try. Each time the dimension selected in the IB algorithm is lower than $n^*$, the corresponding model is trained until convergence. Thus, if we select a range of values from $1$ to $n$ such that
> > $floor(\frac{n-1}{2})$ is lower than $n*$ we will always have at least one full model training, which, for the tested dataset is slower than FONDUE.
> > Furthermore, for any $n \geqslant n* $, the considered models still need to be trained until the mean and sampled representations become stable, that is, for the same number of epochs as
> > FONDUE_IB. Comparing the execution time of FONDUE and the original IB algorithm will thus mostly be based on whether the binary search encounters a situation where $n \geqslant n*$ and require to fully train one or more models or not.
> > - The original IB algorithm will give a value that depends on the range selected. For example, if a good number of dimensions is 30 but the selected range is between 10 and 20,
> > it will return 20. Thus, manually selecting the results could also impact the quality of the predicted number of dimensions. For example one could force the IB algorithm to provide good
> > predictions by selecting a very small range of values consistent with the Elbow methods while a larger range of values would provide worse predictions.
> >
> > We have now clarified this in the paragraph *Why the IB algorithm cannot be directly compared with FONDUE* and for this reason we also run IB with a fixed number of epochs in the
> > supervised context in Appendix K.
> >
> > To summarise, we initially compared our results with the IB metric in the form of FONDUE_IB to avoid biasing the comparison with a specific choice of range for the original IB implementation.
> > We also compared with other existing work which are either based on the Elbow method using reconstruction error or downstream task accuracy in Fig. 5 and 6.
> > We have now added a binary search implementation of the difference between traces for comparison with the original IB algorithm (with model training fixed to a small number of epochs).
> > Overall, we now provide a comparison with all the existing methods presented in the background section except SPI which is an application of the Elbow method aimed at textual data.
> >
> >
> >
> > > This paper purely focuses [...] can be non-Gaussian.
> >
> > This is because the polarised regime has only been studied for these type of models, which represent popular versions of VAEs.
> > However, should a different type of VAE be shown to learn in a polarised regime, one could extend FONDUE to it.
> > An example of such an extension would be to simultaneously apply FONDUE to different layers of a hierarchical VAE with standard Gaussian prior.
> >
> > > The paper claims that [...] existing methods if we do not do a full VAE training.
> >
> > As mentioned above, the metric proposed in Sec. 3 require mean and sampled representations which reflects
> > accurately the number of active and passive variables present in the final model to provide meaningful results.
> > We know from [2,3,4] that the passive, active and mixed variables of the mean and sampled representations are recognisable very early in the training.
> > Thus, we assess the consistency of our results by choosing the number of epochs after which the predicted number of dimensions
> > stop changing as this will occur once the variance of the decoder approaches 0 (see the paragraph *Obtaining stable estimate* in Sec 5.3).
> > The comparison with previous work and the IB algorithm is discussed above.

---

> > > ### Author Response · Authors · 2023-09-06
> > > **Answer to reviewer 1dXF (3/3)**
> > >
> > > References
> > > ----------
> > > [1] Rolinek, M., Zietlow, D., & Martius, G. (2019). Variational autoencoders pursue pca directions (by accident). In Proceedings of the IEEE/CVF Conference on Computer Vision and Pattern Recognition (pp. 12406-12415).
> > >
> > > [2] Dai, B. and Wipf, D. (2018). Diagnosing and Enhancing VAE Models. In International Conference on Learning Representations, vol. 6.
> > >
> > > [3] Dai, B., Wang, Y., Aston, J., Hua, G., & Wipf, D. (2018). Connections with robust PCA and the role of emergent sparsity in variational autoencoder models. The Journal of Machine Learning Research, 19(1), 1573-1614.
> > >
> > > [4] Bonheme, L., & Grzes, M. (2022). How do variational autoencoders learn? insights from representational similarity. arXiv preprint arXiv:2205.08399.
> > >
> > > [5] Boquet, G., Macias, E., Morell, A., Serrano, J. and Vicario, J. L. (2021). Theoretical tuning of the autoencoder bottleneck layer dimension: A mutual information-based algorithm. In 2020 28th European Signal Processing Conference (EUSIPCO), pp. 1512–1516.

---

### Review · Reviewer_zDJy · 2023-08-31

**Summary Of Contributions:**

This work studies the problem of determining the number of latent variables of a VAE. The central goal is to provide an automated method for this task that only requires a short VAE training run (instead of training multiple models to completion). The method is motivated by observations in prior works that some latent variables of a VAE has nearly identical mean and sampled representations (active variables) while some latent variables have a sampled representation with mean $\approx$ 0 and variance $\approx$ 1 (passive variables). The active variables participate in encoding image appearance via the VAE reconstruction loss while the passive variables do not contain sample information but help minimize the KL regularization term during training. The method proposed involves choosing the size of the latent space to match the maximal number of active variables that the model can support, since adding further latent dimensions is believed to provide no additional modeling benefit. In practice, the difference between the trace of the covariance matrix of the mean and sampled representations is used to detect the presence of non-active variables. The number of latent variables is chosen when the difference between these trace values for a partial trained model exceeds a certain threshold. Experiments are conducted to show that the proposed method estimates a number of optimal latent dimensions that is consistent with recent ID estimation methods.

**Audience:**

Yes

**Broader Impact Concerns:**

Broader impacts were not discussed, although this does not impact my assessment of the work.

**Claims And Evidence:**

No

**Requested Changes:**

* The statements of Theorem 1 and 2 must be made much more precise. Currently $\mu$ and $z$ are presented as constant real vectors, in which case their covariance matrices are zero.
* Comparisons with related methods should be made, in particular with IB. I do not find the argument that a fair comparison with IB is not possible to be convincing.
* It should be clarified what the benefit of FONDUE is in terms of the quality of the representation for downstream tasks.

**Strengths And Weaknesses:**

**Strengths**
* Determining the number of latent variables of a VAE is a relevant problem with potential impacts on learning high-quality disentangled representations.
* The direction of using active and passive variables to determine the correct latent size is intuitively reasonable and a novel approach. Given known degeneracy of VAE learning with large latent spaces and many passive variables, selecting a latent space size that bypasses this degeneracy and focuses on the "useful" aspects of VAE learning could be very beneficial for learning more principled probabilistic models, instead of models trained with a probabilistic objective but whose practical learning behavior clearly diverges from the theoretical objective.

**Weaknesses**
* The assumption that, for a given data sample, each dimension of the latent space is essentially either active or passive seems too strong. While it seems quite plausible that many latent dimensions of a VAE can neatly fit into these two categories for a given data sample, I suspect that in practice a mix between active and passive behavior is also quite common (something for example like $\mu_j \approx -0.3$, $\sigma_j^2 \approx 0.7$). The theorems are essentially straightforward consequences of this strong assumption. Since this assumption is so critical, it probably deserves some empirical study on its own.
* The statements of Theorems 1 and 2 are imprecise and do not make sense in their current form. In particular, if both $z$ and $\mu$ are real vectors, their covariance matrix should be 0.
* Essentially no comparison is made to prior works. I do not find the claim that comparison with IB is impossible convincing because this works also sweeps over the number of best dimensions, starting from 1 and going to a large number. To my understanding, both this work and IB either require a reasonable starting range or must use a sweep over a large range without a reasonable starting range. Moreover, methods such as the Information Plane method, which in prior works train full VAEs, should be reimplemented to perform partial training as in this work. The claim that prior works require full training is not convincing unless it can be demonstrated that prior works are not effective with partial training. Overall, better comparison with existing works is essential.
* I do not see any evidence of the benefit of FONDUE for downstream tasks. In Figure 5 and Figure 6, the FONDUE estimate does not correspond to the maximum of the accuracy plot.

---

> ### Author Response · Authors · 2023-09-06
> **Answer to reviewer zDJy (1/3)**
>
> Thank you for your interest in our work and for your detailed comments.
>
> Answers to Reviewer's Questions:
> --------------------------------
> > The assumption that, [...] empirical study on its own.
>
> This behaviour has been shown (empirically and theoretically) to happen in VAEs
> with Gaussian prior with diagonal covariance [1,2,3,4].
> For example, [2,3] assume a decoder such that $p(\mathbf{x}|\mathbf{z})$ is the PDF of $\mathcal{N}(\bar{\mathbf{\mu}}, \lambda I)$ and
> study the phenomenon when $\lambda \to 0$. In the linear case, where the decoder distribution is $\mathcal{N}(W\mathbf{x} + \mathbf{b}, \lambda I)$, given the SVD of
> $W=\boldsymbol{U} \Lambda \boldsymbol{V}^T$ they show that the optimal variance representation is
> $$\boldsymbol{\Sigma} = P \text{diag} \left[ \frac{1}{\frac{\Lambda_{11}^2}{\lambda} + 1}, ..., \frac{1}{\frac{\Lambda_{aa}^2}{\lambda} + 1}, 1, ..., 1 \right] P^T,$$
> where $P$ is a permutation matrix.
> We can directly see that the active variables of the variance representation go to 0 when $\lambda \to 0$. The passive variables are instead equal to 1.
> A similar equality can be derived for the optimal mean representation:
> $$\mathbf{\mu} = P \text{diag}\left[\frac{\Lambda_{11}}{\Lambda_{11}^2 + \lambda}, ..., \frac{\Lambda_{aa}}{\Lambda_{aa}^2 + \lambda}, 0, ..., 0 \right] \boldsymbol{U}^T(\mathbf{x} - \mathbf{b}),$$
> where we can see that the passive variables of the mean representations tend to 0 near the optimal solution.
> They extended these results to the non-linear case using a Taylor expansion near $\mathbf{z} = \mathbf{\mu}$ and found similar results.
> Moreover, they observed that $\gamma$ becomes close to 0 very early in the training, which is a property we exploit to speed up FONDUE.
>
> > The statements of Theorems 1 and 2 [...] matrix should be 0.
> > The statements of Theorem 1 and 2 [...] matrices are zero.
>
> The variance (and covariance) matrices are over the mean and sampled representations obtained for multiple data
> examples (i.e., matrices). We have updated the notation to make this clearer in the paper now.
> We have further updated the definitions, propositions, theorems and related proofs following the same principle as [2,3] and
> re-expressed our results in terms of limits as $\lambda \to 0$ to drop approximations.

---

> > ### Author Response · Authors · 2023-09-06
> > **Answer to reviewer zDJy (2/3)**
> >
> > > Essentially no comparison is made to prior works [...] essential.
> > > Comparisons with related methods [...] IB is not possible to be convincing.
> >
> > We agree that it would be interesting to compare the results obtained with a supervised version with a fixed number of epochs (to avoid biasing the results by fully training some models). We have thus added a comparison between the IB algorithm (trained for a fixed number of epochs sufficient to reach stable results) and the proposed difference between traces combined with a binary search in Appendix K.
> >
> > Regarding the computational time of the IB algorithm, as it is in [5], it is hard to compare with FONDUE as:
> > - The original IB algorithm is based on a manual selection of a range of values to try. Each time the dimension selected in the IB algorithm is lower than $n^*$, the corresponding model is trained until convergence. Thus, if we select a range of values from $1$ to $n$ such that
> > $floor(\frac{n-1}{2})$ is lower than $n^*$ we will always have at least one full model training, which, for the tested dataset is slower than FONDUE.
> > Furthermore, for any $n \geqslant n^* $, the considered models still need to be trained until the mean and sampled representations become stable, that is, for the same number of epochs as FONDUE_IB. Comparing the execution time of FONDUE and the original IB algorithm will thus mostly be based on whether the binary search encounters a situation where $n \geqslant n^*$ and require to fully train one or more models or not.
> > - The original IB algorithm will give a value that depends on the range selected. For example, if a good number of dimensions is 30 but the selected range is between 10 and 20, it will return 20. Thus, manually selecting the results could also impact the quality of the predicted number of dimensions. For example one could force the IB algorithm to provide good predictions by selecting a very small range of values consistent with the Elbow methods while a larger range of values would provide worse predictions.
> >
> > We have now clarified this in the paragraph *Why the IB algorithm cannot be directly compared with FONDUE* and for this reason we also use IB with a fixed number of epochs in the supervised context in Appendix K.
> >
> > To summarise, we initially compared our results with the IB metric in the form of FONDUE_IB to avoid biasing the comparison with a specific choice of range for the original IB implementation. We also compared with other existing work which are either based on the Elbow method using reconstruction error or downstream task accuracy in Fig. 5 and 6. We have now added a binary search implementation of the difference between traces for comparison with the original IB algorithm (with model training fixed to a small number of epochs).
> > Overall, we now provide a comparison with all the existing methods presented in the background section except SPI which is an application of the Elbow method aimed at textual data.
> >
> >
> > > I do not see any evidence of [...] the accuracy plot.
> > > It should be clarified what [...] for downstream tasks.
> >
> > As mentioned in Sec. 2.5, [6] observed that the highest reconstruction and accuracy are generally not obtained for
> > the same number of dimensions. Specifically, a larger number of dimensions will benefit the reconstruction but be
> > detrimental for downstream tasks accuracy.
> > We thus aim to obtain a good trade-off between reconstruction and accuracy, and observed that FONDUE estimates generally
> > locate such point correctly, ignoring diminishing returns for more complex datasets like Figure 6c, as discussed in Sec.5.2 .
> > Furthermore, as discussed in Sec. 1, [7] observed that the passive variables of mean representations (which are used for downstream tasks) were highly correlated with multiple active variables which could be detrimental to the quality and interpretability of downstream task models which are sensitive to correlated inputs (e.g., linear regression).

---

> > > ### Author Response · Authors · 2023-09-06
> > > **Answer to reviewer zDJy (3/3)**
> > >
> > > References
> > > ----------
> > > [1] Rolinek, M., Zietlow, D., & Martius, G. (2019). Variational autoencoders pursue pca directions (by accident). In Proceedings of the IEEE/CVF Conference on Computer Vision and Pattern Recognition (pp. 12406-12415).
> > >
> > > [2] Dai, B. and Wipf, D. (2018). Diagnosing and Enhancing VAE Models. In International Conference on Learning Representations, vol. 6.
> > >
> > > [3] Dai, B., Wang, Y., Aston, J., Hua, G., & Wipf, D. (2018). Connections with robust PCA and the role of emergent sparsity in variational autoencoder models. The Journal of Machine Learning Research, 19(1), 1573-1614.
> > >
> > > [4] Lisa Bonheme and Marek Grzes. The polarised regime of identifiable variational autoencoders. ICLR
> > > TinyPapers, 2023.
> > >
> > > [5] Shujian Yu and José C. Príncipe. Understanding autoencoders with information theoretic concepts. Neural
> > > Networks, 117:104–123, 2019. ISSN 0893-6080.
> > >
> > > [6] Kien Mai Ngoc and Myunggwon Hwang. Finding the best k for the dimension of the latent space in
> > > autoencoders. In Computational Collective Intelligence, pp. 453–464. Springer International Publishing,
> > > 2020. ISBN 978-3-030-63007-2.
> > >
> > > [7] Lisa Bonheme and Marek Grzes. Be More Active! Understanding the Differences between Mean and Sampled
> > > Representations of Variational Autoencoders. arXiv e-prints, 2021.

---

### Author Response · Authors · 2023-09-06
**Updates to the paper**

Dear reviewers,
Thank you for your interest, time and effort.

We are happy to see that our proposed algorithm was found appealing (reviewer 6PWv), easy to understand and to
implement (reviewer 1dXF) and potentially beneficial to learn more principled probabilistic models (reviewer zDJy).


Major updates
==============
There were two main concerns amongst the reviewers:
1) The theorems, propositions, and proofs lack precision due to the approximation sign (reviewers 6PWv and 1dXF),
and unclear notation of several terms (reviewers zDJy and 1dXF).
2) No comparison is performed with the original (supervised) IB algorithm (reviewers 1dXF and zDJy).

Updates on the theoretical part
--------------------------------

To address the first point, reviewer 1dXF recommended removing all approximations, which we have now done by re-expressing
all the mathematical statements in terms of limits as the decoder's variance tends to 0, in a similar way to [1,2]. As the
decoder's variance drops very early in the training (as per [1]), this does not impact our assumption that active, passive and
mixed variables can be distinguished after a few epochs, which we have emphasised in the updated version of the paper.
See our answer to reviewer 1dXF for a more extensive discussion on that topic.

Reviewer 6PWv and zDJy also recommended better defining the different terms used (e.g., mean and sampled representations,
$\epsilon$, $Var[\mu]$) as some were not introduced or their dimensionality was unclear. We have now addressed this and
clarified our notation (see related answers to reviewers 6PWv and zDJy for more details).

Updates on the experimental part
--------------------------------

Reviewers 1dXF and zDJy found the given explanations about not using the original IB algorithm insufficient and requested
additional comparison in the supervised context.
We have addressed this issue in two ways:
- We now compare the IB algorithm and Thm. 3 in the supervised context using binary search
(for IB, this is equivalent to the original implementation except that we always stop after a few epochs).
For the chosen range of values (1 to 200) we observed that the binary search was slower in both cases as
it generally needed a few more iterations to terminate (and thus more partial model training), except for Symsol where
the number of iterations (and thus execution time) is similar.
As expected the predicted number of dimensions remained consistent between supervised and unsupervised settings with
Thm. 3 being closer to the Elbow methods than IB. The results can be found in App. K.

- We have added a more thorough justification of why we did not use the supervised IB algorithm in the first place in
the paragraph *Why the IB algorithm cannot be directly compared with FONDUE*, showing how we could either argue that IB
is faster or slower than FONDUE for different carefully selected ranges of values.


Minor updates
=============

- We have added new experiments with different initialisation (see App.H), learning objectives and hyperparameters
(see paragraph *Can FONDUE be applied to other architectures, hyperparameters and learning objectives?* and App. L)
to display the generalisability of FONDUE as suggested by reviewer 6PWv.
- Any non-essential content in the related work section was moved to App. J to improve readability as proposed by reviewer 6PWv.
- We have clarified the contribution of our work with respect to previous work as recommended by reviewer 1dXF.
- We have reformulated several unclear sentences reported by reviewer 6PWv.


To ease the reviewing process, we have uploaded a diff file in supplementary material to highlight all the changes to
the previous version of the paper in (we hope) a transparent and clear manner.

We hope that these updates clarify the paper and alleviate any concerns you had.
We are happy to answer any other questions you may have.



References
----------
[1] Dai, B. and Wipf, D. (2018). Diagnosing and Enhancing VAE Models. In International Conference on Learning Representations, vol. 6.

[2] Dai, B., Wang, Y., Aston, J., Hua, G., & Wipf, D. (2018). Connections with robust PCA and the role of emergent sparsity in variational autoencoder models. The Journal of Machine Learning Research, 19(1), 1573-1614.

---

### Decision · Action_Editor_17gT · 2023-11-09

**Recommendation:** Reject

**Comment:**

The review process for this paper was very delayed, and for my part in that I apologize.  We had difficulty acquiring enough reviewers and in the end even had difficulty extracting final recommendations from the final reviewer.

I want to thank the reviewers for their participation and reading and review of the paper, and I want to thank the authors for being so responsive to the reviewers.  I do believe the paper has been improved given the updates, but unfortunately, I cannot, at this time support the paper for acceptance.

**Audience:**

The paper discusses the problem of determining the proper latent dimensionality for VAEs, which clearly has an audience at TMLR.

**Claims And Evidence:**

As with the reviewers, despite positive changes made by the authors, I'm concerned that the claims being made in the paper are overstated given they level of mathematical and experimental support.

I agree with Reviewer zDJy that the paper would be improved it the claims were softened, away from being statements that read as mathematical certainty and the paper instead modified to largely be reduced claims about the performance of the proposed algorithm supported by empirical evidence.

At this time, I need to vote to reject the paper.

---

> ### Author Response · Authors · 2023-11-14
> **Further questions to improve our paper**
>
> Dear action editor and reviewers,
>
> Thank you for your helpful and constructive comments.
>
> To further improve our paper, we would be very grateful if the action editor could please provide further details on some points of the final decision.
> Specifically, we are confused by the statement: "the claims being made in the paper are overstated given they level of mathematical and experimental support".
>
> To address this issue, we updated the paper as follows during the discussion period (see the "Updates to the paper" for more details):
> - As suggested by reviewer 1dXF, we rewrote our theorems and proof based on the work of [1,2] instead of the definition of [3] to remove any approximation.
> - We also added two new experiments for better comparison with the IB algorithm upon the request of reviewer 1dXF and zDJy.
>
> We would be grateful if the action editor could please provide more details on why these updates do not address the reviewers' concerns. Indeed, a better understanding of this issue would greatly help us to improve the quality of our paper.
>
> Thank you for your help,
>
> The authors.
>
> References
> =========
>
> [1] Dai, B. and Wipf, D. (2018). Diagnosing and Enhancing VAE Models. In International Conference on Learning Representations, vol. 6.
>
> [2] Dai, B., Wang, Y., Aston, J., Hua, G., & Wipf, D. (2018). Connections with robust PCA and the role of emergent sparsity in variational autoencoder models. The Journal of Machine Learning Research, 19(1), 1573-1614.
>
> [3] Rolinek, M., Zietlow, D., & Martius, G. (2019). Variational autoencoders pursue pca directions (by accident). In Proceedings of the IEEE/CVF Conference on Computer Vision and Pattern Recognition (pp. 12406-12415)

---

> > ### Comment · Action_Editors · 2024-01-12
> > **Further Comments**
> >
> > Sorry for the delay, I was on break from the holidays.
> >
> > I'm also sorry in the sense that I don't think the process for this paper was in keeping with the spirit of TMLR, or its desire of 'quick turnaround'. It was difficult to recruit reviewers and solicit responses from them all, and the process was delayed as a result.  I can understand that it is frustrating as authors to have a paper in limbo for a long time and I'm sorry for my part for not trying harder to move things along quicker.
> >
> > I've taken a closer look at the paper and the reviews and discussion to date and I stand by the decision to recommend rejection.  I do not think the paper is appropriate for TMLR. I'm concerned about both the accuracy and potential interest of the paper as written.
> >
> > Overall, the paper suggests that gaussian VAEs operate in a "polarizing" regime as suggested by previous work, wherein every latent variable is either active, passive or potentially mixed.  The active variables are the ones that concentrate on some nonzero mean as the decoder variance vanishes, the passive ones are the ones that concentrate on the prior (i.e. independent gaussians, typically with zero mean and unit variance), and the mixed ones are in between.  The paper strongly assumes that gaussian vaes all have this property, and furthermore that if we vary the dimensionality of the hidden representation, a VAE will preferentially learn active, then mixed, then passive latent variables.  With all of these assumptions, the paper suggests computing the difference of the trace of the variance of the stochastic activations and their means as a metric to tract the number of 'active' components.  Active variables contribute nothing, passive ones contribute one per dimension and mixed are in between.  The paper then presents an algorithm that essentially does a sort of bounded interval search to try to find the hidden dimension size for which the value this metric takes is below some threshold (usually taken to be 1).  Furthermore, the paper claims that this metric settles in early in training so that the can do this discrimination without training the VAEs to completion.
> >
> > Concerning accuracy, I, like reviewer zDJy do not like the strong assumptions underlying the whole work, that gaussian VAEs have this strict assignment into three classes of hidden variables, nor do I believe that we can expect or that either previous work or this work demonstrates that even if all of the hidden dimensions could be so characterized that we should expect the order in which they are learned to always be active -> mixed -> passive.  That assumption underlies the whole of the work.  Beyond that I take some issues that the work for the most part is written as though it pertains to all VAEs while its very specific to only gaussian mean field VAEs, a very restricted set, with comparably less general interest.
> >
> > I fear the paper conflates compression with geometry.  In the introduction, the paper says that training high dimensional latents "...  [defeat] the purpose of learning compressed representations...".  This is simply untrue.  Even high dimensional representations can be highly compressed if they do not differ from the prior and have low rate.  In the language of this paper, even a 1000 dimensional representation, if all of the dimensions were passive is highly compressed in the information theoretic sense.  I fear this conflation underlies more or less the whole of the paper.
> >
> > Overall, the paper grossly oversells itself.  At its core, there is a simple idea here, that one could try to use the difference between the stochastic and mean representations as a heuristic to track and set the appropriate dimensionality of a gaussian VAE.  There is utility in that idea, but we have to be aware of its limited scope and reliance on strong assumptions.  For a TMLR audience, if the paper earnestly presented this as a heuristic and demonstrated its empirical utility in cases I think it would be a fine candidate for acceptance.  Instead, the paper has fallen for the trap that what makes for a good machine learning paper is having Prepositions, Theorems, Algorithms and Proofs even if of little consequence.  Unfortunately, this presentation only obfuscates the scientific discussion to the point of being misleading and inaccurate and no longer provides generalizable insights to the TMLR audience.  I have to stand by my recommendation for rejection.

---

> > > ### Author Response · Authors · 2024-01-30
> > > **Thank you for your detailed answer (1/2)**
> > >
> > > Dear action editor,
> > >
> > > Thank you for your time and detailed answers. Please note that we have been patient since the initial submission of this paper. However, the fact that there was no discussion period with the reviewers has led to misunderstandings, which we could have easily clarified. We will make another attempt below.
> > >
> > > ## Polarised regime
> > >
> > > We would like to emphasise that the order in which passive and active variables are learned is not an additional assumption but truly a consequence of the polarised regime. Let us consider a linear Gaussian VAE with diagonal covariance.
> > > As per [1,2,3], the ELBO can be simplified to
> > > $$- \Big( \sum_{i=1}^a \log \big(\frac{1}{\lambda} \Lambda_{ii}^2 + 1 \big) + m \log \lambda \Big) - (\mathbf{x}-\mathbf{b})^T(U \Lambda^2 U^T + \lambda I_m)^{-1}(\mathbf{x}-\mathbf{b})$$.
> > > Here $\Lambda$ is the diagonal matrix containing the singular values of the decoder's weights, $a$ is the number of non-zero singular values, $U$ is a unitary matrix, $\mathbf{b}$ a bias term and $\lambda$ the scaling term of the decoder's variance.
> > >
> > > We can see from this equation that there is some pressure to reduce the number of non-zero singular values while maintaining a good reconstruction quality. This number of non-zero singular values, $a$, will correspond to the number of active variables as, if we compute the MLE of the encoder variance and mean, we obtain
> > > $$\Sigma = P \text{diag}\Big[\frac{1}{\frac{\Lambda^2_{11}}{\lambda}+1}, \cdots, \frac{1}{\frac{\Lambda^2_{aa}}{\lambda}+1}, 1, \cdots, 1\Big] P^T$$
> > > and
> > > $$\mathbf{\mu} = P \text{diag}\Big[\frac{\Lambda_{11}}{\Lambda_{11}^2 + \lambda}, \cdots, \frac{\Lambda_{aa}}{\Lambda_{aa}^2 + \lambda}, 0, \cdots, 0\Big] U^T(\mathbf{x}-\mathbf{b})$$, where $P$ is a permutation matrix.
> > >
> > > Once one gets passive variables (i.e., once we have more than $a$ dimensions), these are superfluous dimensions that the decoder does not need for reconstruction.
> > > If one were to increase the number of dimensions further, the new variables could only be passive as additional active variables increase the KL divergence without a substantial gain in reconstruction quality.
> > > Dai and Wipf have proved the existence of this phenomenon for the non-linear case, and a similar discussion in that context can be found in p.7-8 of [3]. Multiple studies have further observed the early convergence of the different variable types (e.g., [2,3]).
> > >
> > > ## Compression
> > >
> > > Regarding compression, we meant learning representations that are compressed yet meaningful. In other words, latent representations are obtained by models with a good rate-distortion trade-off.
> > > In your example, one would have posterior collapse (all variables passive). While the rate would be low, the distortion would be high (i.e., poor reconstruction), as it was nicely illustrated in Fig. 1 of [4].
> > > That is why we emphasised that we seek a trade-off between reconstruction and regularisation and discussed ELBOW methods.
> > > Thank you for pointing out that this was unclear. We will reformulate it in our subsequent iterations to avoid any future misunderstandings.
> > >
> > > ## The need for formal notation
> > >
> > > Please note that zDJy did not ask for less formal notation, and they asked us to cope with $\approx$, which we did. Additionally, given the formal nature of the core papers that we built our paper upon, we felt that the current level of technical exposition would be appropriate. If we should read the final comments in your previous post as a request to add a more informal and intuitive presentation of the polarised regime and the proposed algorithm, then we would be happy to provide such additions. It is unfortunate that we did not get the opportunity to hear this feedback during the discussion period, as we would have been happy to make such changes in the previous revision.
> > >
> > > We thank you again for your time and valuable comments, which will contribute to improving this paper. However, we still believe this paper was rejected without following the procedures of the TMLR. For example, please note that the reviewer guidelines https://jmlr.org/tmlr/reviewer-guide.html mention this: "You are expected to actively engage with the authors until the issues you have raised for discussion have been sufficiently explored." in reviewing responsibilities. This was really the reason why we wanted to submit to TMLR, but our paper has now been rejected without allowing us to have a discussion with the reviewers. They did not engage, and there was no discussion.

---

> ### Author Response · Authors · 2024-01-30
> **Thank you for your detailed answer (2/2)**
>
> ## References
>
> [1] Lucas, J., Tucker, G., Grosse, R. B. and Norouzi, M. (2019a). Don't Blame the ELBO! A Linear VAE Perspective on Posterior Collapse. In Advances in Neural Information Processing Systems, vol. 32
>
> [2] Rolinek, M., Zietlow, D. and Martius, G. (2019). Variational Autoencoders Pursue PCA Directions (by Accident). In Proceedings of the IEEE/CVF Conference on Computer Vision and Pattern Recognition (CVPR).
>
> [3] Dai, B. and Wipf, D. (2018). Diagnosing and Enhancing VAE Models. In International Conference on Learning Representations, vol. 6.
>
> [4] Alemi, A. A., Poole, B., Fischer, I., Dillon, J. V., Saurous, R. A. and Murphy, K. (2018). Fixing a Broken ELBO. In Proceedings of the 35th International Conference on Machine Learning, ICML 2018, Proceedings of Machine Learning Research, vol. 80.